

# Nitrogen and phosphorus recycling mediated by copepods in Western Tropical South Pacific

Valentina Valdés[1,2], François Carlotti[3], Ruben Escribano[4,5], Katty Donoso[3], Marc Pagano[3], Veronica Molina[6] and Camila Fernandez[2,7,8]

[1]Programa de Doctorado en Oceanografía, Departamento de Oceanografía, Facultad de Ciencias Naturales y Oceanográficas, Universidad de Concepción, Concepción, Chile.
[2]UPMC Univ Paris 06, UMR 7621, Laboratoire d'Océanographie Microbienne, Observatoire Océanologique, Sorbonne Universités, Banyuls sur Mer F-66650, France.
[3]Aix-Marseille Univ, Universite de Toulon, CNRS, IRD, OSU PYTHEAS, Mediterranean Institute of Oceanography MIO,
UM 110, 13288, Marseille, Cedex 09, France.
[4]Departamento de Oceanografía, Universidad de Concepción, Concepción, Chile.
[5]Instituto Milenio de Oceanografía, Universidad de Concepción, Concepción, Chile.
[6]Departamento de Biología and Programa de Biodiversidad, Facultad de Ciencias Naturales y Exactas, Universidad de Playa Ancha, Playa Ancha, Valparaíso, Chile.
[7]FONDAP INCAR center (15110027), and PFB-31 COPAS Sur Austral program; Departamento de Oceanografía, Universidad de Concepción, Concepción, Chile.
[8]Centro FONDAP de Investigación en Dinámica de Ecosistemas Marinos de Altas Latitudes (IDEAL), Valdivia, Chile.

*Correspondence to*: Valentina Valdés (vvaldesc@udec.cl)

**Abstract.** Zooplankton play a key role in the regeneration of nitrogen and phosphorus in the ocean through grazing and metabolism. This study investigates the role of the organic and inorganic nitrogen and phosphorus compounds released by copepods on biogeochemical processes and on the microbial community composition during the OUTPACE cruise (18 February – 3 April 2015) at three long duration stations (LD). Two LD stations were located in the Melanesian Archipelago region (MA; LD A and LD B) and one in the South Pacific Gyre (SG; LD C), which represent oligotrophic and ultra-
oligotrophic regions respectively. At each station, microcosm onboard experiments were performed with locally sampled organisms, comprising a mix of epipelagic copepods fed with their natural food and then incubated along with wild microbial assemblages. In presence of copepods, ammonium and dissolved organic nitrogen showed a significant increase, compared to a control in two situations: in ammonium concentration (increasing rate: 0.29 µmol L⁻¹ h⁻¹ after 4 h of incubation) in LD C and in dissolved organic nitrogen concentration (rate: 2.13 µmol L⁻¹ h⁻¹ after 0.5 h of incubation) in LD
30   A. In addition, during the three experiments, an enhanced remineralization (ammonification and nitrification) was observed when adding copepods compared to the controls. A shift in the composition of active bacterial community was observed for the experiments in LD A and LD B mainly characterized by an increase in Alteromonadales and SAR11, respectively and linked with changes in nutrient concentrations. In the experiment performed in LD C, both groups increased but at different periods of incubation, Alteromonadales between 1 and 2 h after the beginning of the experiment, and SAR 11 at the end of
incubation. Finally, our experimental results in near in situ conditions, show that copepods can be a source of organic and




inorganic compounds for bacterial communities, which respond to excretion pulses at different scales, depending on the initial environmental conditions and on their community composition. These processes can contribute significantly to nutrient recycling in the epipelagic ecosystem of ultra and oligotrophic oceanic regions.

## 1 Introduction

The Tropical South Pacific is considered one of the most oligotrophic regions in the world ocean. Its biogeochemical characteristics include nitrogen and phosphate limitation which modulate the low biological productivity (Moutin et al., 2008). In addition, the strong thermal stratification, enhanced by current global warming, decreases the nutrient supply to the euphotic zone. The Western Tropical South Pacific (WTSP) was recently recognized as a hot spot of N2 fixation (Bonnet et al., 2017) and the input of new nitrogen to the surface ocean through this process sustains over 50 % of the primary
productivity (Carpenter et al., 2004; Karl et al., 1997). However, it has been reported that predator-prey interactions can provide substantial amounts of nitrogen and phosphorus, and supply an alternative substrate for phytoplankton and heterotrophic microorganisms stimulating the microbial loop in a variety of areas, contributing to the regenerated and new production (Richardot et al., 2001; Vargas et al., 2007; Arístegui et al., 2014; Valdés et al., 2017).

Metazooplankton metabolism is recognized as one of the main mechanisms controlling concentration, composition and cycling of dissolved organic matter (DOM) in the sea (Sipler and Bronk 2014). Zooplankton may release dissolved organic and inorganic matter through sloppy feeding (Møller, 2004, 2007), leaching of fecal pellets (Hasegawa et al., 2000; Steinberg et al., 2002) and excretion (Saba et al., 2011). Along with bacterial remineralization, it determines the amount of regenerated nitrogen and phosphorus available for phytoplankton production (Steinberg and Landry, 2017). Copepods
excrete up to 53% of its body nitrogen per day, mainly in the form of ammonium, thereby recycling much of the nitrogen in the water column (Steinberg and Saba 2008). Reports also indicate that zooplankton could sustain between 40-50% of the nitrogen requirements of phytoplankton in open ocean areas as North Pacific central gyre (Eppley et al., 1973). Reports indicate that diel vertical migrating zooplankton can sustain between 2 to 19% of the nitrogen requirements by phytoplankton through ammonium excretion at the oligotrophic Sargasso Sea (Steinberg et al., 2002). Furthermore,
zooplankton can also excrete substantial amounts of dissolved organic nitrogen (DON), i.e., between 7 to 80% of the total dissolved nitrogen released (Steinberg et al., 2002). In fact, the composition and quantity of the compounds excreted are highly dependent on the type of food ingested by zooplankton (Miller and Roman, 2008; Saba et al., 2009). Even less studied than nitrogen is the phosphorus release by zooplankton. It has been reported, that one-third of the phosphorus released by zooplankton (amphipods) was in the organic form (Johannes, 1964; Satomi and Pomeroy, 1965). Isla et al.
(2004) estimated that mesozooplankton phosphate excretion can sustain 21.7 % of phytoplankton phosphate requirements in oceanic water of the NW Iberian. However, the impact of zooplankton metabolism on the phosphorus recycling is largely



unknown in oceanic areas. Titelman et al. (2008) demonstrated that copepod feeding enhance organic phosphorus uptake by bacterial activity in phosphorus limited system.

The importance of quality and quantity of DOM in structuring bacterioplankton community has been increasingly reported

(Alonso-Saez and Gasol 2007). However, the processes and mechanisms through which nitrogen and phosphorus compounds released by zooplankton can determine and influence the food web and the structure of the microbial community are unclear. Recent research has demonstrated that bacterial community composition can change in response to the addition of DOM (Landa et al., 2013; Sarmento et al., 2013). Studies on zooplankton-produced DOM by bacterial communities are scarce, however, recent reports demonstrated that the addition of excretory products released by zooplankton (krill)

stimulates bacterial growth and production in the Southern Ocean (Arístegui et al., 2014). Specific studies carried out by us in a eutrophic coastal zone off southern/central Chile indicate that the nitrogen excreted by copepods generate a specific response of nitrifying communities and in the active marine bacterioplankton community (Valdés et al., 2017a, b). However, studies of copepod nitrogen and phosphorus excretion impact on microbial community from oligotrophic and ultra-oligotrophic areas can provide valuable information about of the recycling of this nutrients in larger areas of the ocean.

Herein, we studied the role of organic and inorganic dissolved compounds released by copepods along an oligotrophic gradient in the WTSP and its potential effect on their recycling and structuring of the bacterioplankton community.

## 2 Material and methods

### 2.1 Study area and sampling strategy

Sampling was conducted in Western Tropical South Pacific during austral summer 2015 (18 February -3 April), on board the RV L`Atalante. The transect beginning in west of New Caledonia (18º S-159.9º E) and ending near Tahiti (17.56º S-149.05º W). Experiments were conducted in three long duration stations: LD A, LD B and LD C, which are located in two geographic regions: LD A and LD B in the Melanesian archipelago (MA) and LD C in the subtropical gyre (SG). These chosen stations based on the sea surface chlorophyll-a concentration, which was maximum in LD A and LD B, and

minimum in LD C station. At each sampling station seawater samples were obtained using Niskin bottles (12 L) arranged on a CTD rosette from the Deep Chlorophyll Maximum (DCM). Seawater was collected for chemical and biological initial characterization, including ammonium ($NH_4^+$), nitrate ($NO_3^-$), nitrite ($NO_2^-$), phosphate ($PO_4^{3-}$), DNA and RNA. The DCM depths correspond to 80 m, 34 m and 140 m in LD A, LD B and LD C, respectively.





### 2.2 Mesozooplankton sampling

Live zooplankton samples were captured between 0 to 100 m depth under night conditions in the three LD stations using a Bongo zooplankton net, with 120 μm mesh size, equipped with a non-filtering cod-end to obtain undamaged individuals. Live samples were immediately transferred to coolers until sorting at the laboratory on board.

### 2.3 Preparation of the microcosm and experimental setup

The experiment design consists in three steps: (1) Copepod acclimation, (2) feeding and (3) copepod+microbial recycling (Fig. 1). Seawater for incubations (30 L) was collected into clean (10% HCl rinsed) polycarbonate carboys. For acclimatizing (2) and copepod+microbial recycling (3) steps, seawater (22.5 L) was immediately filtered onto a 0.7 μm (GF/F; Whatman) using a peristaltic pump. The remaining 7.5 L were used in the feeding phase. Seawater for the different

steps was maintained in a cold room (in situ temperature ~25º C) until the beginning of the experiment.

Undamaged individuals were sorted from the alive samples and identified using a stereomicroscope at low light. Copepods samples in the three experiments consisted in a mix of the most representative copepods (adults) in the sample (Table S1). In the acclimatization phase (1), 15 groups of 10 copepods were incubated in 500 mL Nalgene bottles, maintained for 4-6 h in

filtered seawater previously filtered (GF/F Whatman) in darkness and at controlled temperature (in situ). In the feeding step (2) copepods were removed from the bottles using a sieve (20 μm) and maintained in new 500 mL polycarbonate bottles (Nalgene) with <150 μm filtered seawater (polycarbonate membrane) for feeding. This step lasted 4 hours and, as the acclimatization phase, was under controlled temperature. Thereafter, each copepod group (15 groups) was placed in a new set of 500 mL polycarbonate bottles (Nalgene) filled with the seawater previously filtered (0.7 μm; Millipore) for the

excretion phase. Also, other 15 bottles without copepods were incubated as control. In total, 30 bottles were used in this step at LDA, whereas additional bottles were included at LD B and C, such that 6 bottles were added in that case. The seawater used was filtered to avoid the presence of phytoplankton and small protists, to minimize grazing effect and ensure the presence of natural microbial assemblages. At the end of each step copepods were observed and checked by direct observation of ensuring swimming behaviour in the bottles.

At time intervals of 0 h (T0), 0.5h (T1: T0+0.5 h), 1 h (T2: T0+1 h), 2 h (T3: T0+2h) and 4 h (T4: T0+4 h) samples were collected for $NH_4^+$, $NO_3^-$, $NO_2^-$, $PO_4^{3-}$, DON, dissolved organic phosphorus (DOP), bacterioplankton abundance and RNA. An additional sampling, corresponding to incubation time of 15 min (T0´: T0+0.25h), was added just for having more measurements of $NH_4^+$ in the experiments of stations LD B and LD C. Six bottles were sacrificed at each time, three for

treatment with copepods and three for control (without copepods). The three bottles sacrificed at each time represented triplicates for treatment and control.



At the end of each observation time, copepod samples were preserved immediately with formalin buffered by sodium borate (5% final concentration) for further zooplankton identification analysis (Table S1).

**2.4 Dissolved inorganic and organic nutrients measurements**

Nutrient samples ($NO_3^-$, $NO_2^-$ and $PO_4^{3-}$) were taken in duplicate in 20 mL HDPE bottles, filtered through 0.7 µm filters (GF/F; Whatman) and poisoned with $HgCl_2$ to a final concentration of 20 µg $L^{-1}$. Samples were stored at -20 °C and analyzed on board determined by standard colorimetric techniques (Aminot and Kérouel 2007), using a SEAL Analytical AA3 HR system (SEAL Analytical, Serblabo Technologies, Entraigues Sur La Sorgue, France). Samples for $NH_4^+$ determination (40 ml) were taken in duplicate and analyzed on board using a Jasco FP-2020 fluorometer according to

Holmes et al., (1999). Samples for DON and DOP (30 mL) were filtered through pre-combusted (450º C, 6h) GF/F filters (Whatman). Samples were collected in Teflon bottles and analyzed immediately on board by the wet oxidation method (Pujo-Pay and Raimbault, 1994). DON and DOP concentrations were determined by sample oxidation (30 min, 120º C) and corrected for $NO_3^-$, $NO_2^-$ and $PO_4^{3-}$ concentrations, respectively.

**2.5 Bacterioplankton abundance, DNA and RNA extraction**

Bacterioplankton abundance was determined by flow cytometry. From each bottle 1,350 µL samples were fixed in sterile cryovials with glutaraldehyde (at 0.1% final concentration). The samples were stored at -80º C until laboratory analysis. Samples were analyzed by flow cytometry (FACScan, Becton Dickinson). The abundance of non-fluorescent picoplankton was estimated from samples previously stained with SYBR green I (Molecular probes) according Marie et al. (1997). DNA samples for the initial characterization of bacterial communities were collected at DCM depth of each sampling site. Samples

(9-10 L) were filtered through cellulose ester filters (0.22 µm; Millipore) using a peristaltic pump and stored with RNA later reagent (Ambion) at -20º C until extraction procedures.

DNA was isolated u sing PowerSoil DNA Isolation Kit (MoBio Laboratories) in accordance with the manufacturer´s specifications following Levipan et al. (2014). DNA was quantified by spectrophotometry (NanoDrop ND-1000

Spectrophotometer). RNA samples were collected for initial characterization of active bacterial community at DCM at each LD station and during each experiment to study the active bacterial community composition. Seawater (100 mL) was filtered using a sterilized syringe and 25 mm swinnex through 0.22 µm hydrophilic PVDF filters (Millipore) and the filters were preserved with RNAlater solution (Ambion) and stored at -80º C until RNA extraction procedures.

RNA samples were extracted using Mirvana kit (AM1560; Ambion) in accordance with manufacturer´s instructions including a mechanical disruption step and homogenization using 200 µm diameter zirconium beads (Low Binding Zirconium Beads, OPS Diagnostic) and homogenized twice at ~3,000 rpm for 30 s by using Mini-Beadbeater-8TM (Biospec



Products). In order to remove DNA traces, RNA was treated with TURBO DNA-free kit (Ambion). Finally, concentration and quality (A260/A280 ratio) of RNA extracts as well with DNA extracts were determined by spectrophotometry (NanoDrop ND-1000 Spectrophotometer).

**2.6 Bacterial community structure analysis**

5   Bacterial community structure was analyzed by Illumina MiSeq sequencing method from 16S rRNA (samples from incubation), using cDNA as template, and 16S rDNA in situ, following Campbell and Kirchman, (2013). cDNA was generated using random primers provided by the ImProm-IITM Reverse Transcription System (Promega). Bacterial 16S rRNA gene libraries from V1-V3 region (27F-519R) were generated at the Molecular Research LP (www.mrdnalab.com, Shallowater, TX, USA).

The 16S rRNA and rDNA gene sequences were processed using Mothur software v1.35.1 (Schloss et al., 2009). Sequencing data sets were curated by quality filtration to minimize the effects of random sequencing errors by eliminating sequence reads <200 bp and trimming of sequences that contained more than one undetermined nucleotide (N) and sequences with a maximum homopolymer length of 8 nucleotides. Chimeric sequences were identified using the Chimera UCHIME algorithm

(Edgar et al., 2011) and removed to retain high quality reads.

The 16S rRNA and rDNA gene sequences retrieved were taxonomically classified using the automatic software pipeline SILVAngs available from https://www.arb-silva.de/ (Quast et al. 2013). In LD A station, a total of 178,097 sequences were analyzed, ranging between 10,831 and 19,703 sequences for each library (Fig. S1). In LD B station 219,019 sequences were

analyzed from 15,260 to 20,966 for each library, and in LD C 230,805 sequences were analyzed, ranging between 13,699 and 22,104 sequences for each library. In each of the three LD stations, the highest number of sequences correspond to DNA in situ. Libraries were deposited in the European Nucleotide Archive (ENA) under study accession PRJEB21648 with the following run access numbers: ERS1810581-ERS1810616.

The composition was analyzed at Phyla, Class and Order taxonomic level as abundant (>0.5%) and others (<0.5%) in total sequences retrieved from each library. The Chao, Shannon and Evenness indices were calculated using Past3 software. Previously, since the number of sequences per sample was variable, we normalized the different libraries sizes by subsampling routine in Mothur software version 1.36 (Schloss et al., 2009), to reflect the lowest number of sequences encountered (LD A: 10,831; LD B: 15,260 and LD C: 13,699).

**2.7 Statistical analysis**

Statistical analysis of treatments effects on chemical and biological parameters were performed using a two-way analysis of variance (ANOVA) after checking normality assumptions (Kolmogorov-Smirnov test) and homoscedasticity (Levene´s test).




Pairwise multiple comparisons were performed using a Tukey test as a posteriori analysis. Statistical significance was set a p=0.05 and analysis was computed using R software. Time point sampled at 15 min was not consider for this analysis, because of too few data.

5    Bacterial community structure changes were compared using ordination Bray-Curtis similarities and used to build dendrograms by the unweighted pair group method with arithmetic averages (UPMG). For multivariate statistical analysis, the software package PRIMER v.6 and the add-on PERMANOVA+ was used. The permutational multivariate analysis of variance (PERMANOVA) with fixed factor was applied to investigate the differences of bacterial community composition for treatments and control in both experiments. Principal co-ordinate analysis (PCO) was performed to visualize patterns of 10    the bacterial community in response to the different treatments.

### 3 Results

#### 3.1 Biogeochemical in situ conditions

Inorganic nutrients concentrations, obtained at the DCM depth, varied substantially between the sampling sites (Table 1). Lower concentrations close to the limit of detection of the methods of $NH_4^+$, $NO_3^-$, $NO_2^-$ and $PO_4^{3-}$ were observed in LD B 15    (0.01, 0.05, 0.02 and 0.03 µmol $L^{-1}$, respectively), compared to LDA and LD C. The highest concentrations were observed in LD A, particularly for $NO_3^-$ (1.53 µmol $L^{-1}$). The resulting N: P ratio showed greater values in LD A (9.97 ± 0.70) than LD B and LD C, and the lowest N: P ratio (1.21 ± 0.11) was estimated for LD C seawater.

#### 3.2 Changes in inorganic nutrients, DON and DOP during the incubations

Inorganic nitrogen and phosphorus concentrations changed substantially over the course of the experiment at the three LD 20    stations (Fig. 2 and 3). Ammonium concentrations (Fig. 2a, e, i) increased significantly (four- and seven-fold) in the presence of copepods in all experiments (LD A: $F_{1,20}$: 93.2, p<0.001; LD B: $F_{1,18}$: 61.7, p<0.001; LD C: $F_{1,20}$: 108.9, p<0.001). The largest and most significant difference in ammonium concentration with respect to the controls was associated with T0 in LD A, T3 in LD B and T4 in LD C (ANOVA and Tukey`s test p<0.001; Table S2). Nitrate showed significant differences between the treatment with copepods and controls in LD B and LD C; characterized by notorious accumulation 25    at the first treatment (Fig. 2f, j) at T1 in LD B (ANOVA and Tukey`s test p=0.02; Table S2), and at T4 in LD C experiments (ANOVA and Tukey`s test p=0.009; Table S2). Nitrite (Fig. 2c, g, k) were slightly variable through time in the experiments and no significant difference was detected between copepods and the controls (LD A: $F_{1,16}$: 0.50, p=0.49; LD B: $F_{1,18}$: 0.13, p=0.7; LD C: $F_{1,20}$: 0.44, p=0.51). DON (Fig. 2d, h, l) showed significant differences between the treatment and control in LD A and LD C (LD A: $F_{1,20}$: 8.99, p=0.007; LD C: $F_{1,20}$: 29.0, p<0.0001), and this difference was associated with the 30    increment in concentration at T1 in LD A and T0 in LD C in the treatment with copepods. At LD B non-significant differences were detected between treatments and controls trough the time (ANOVA and Tukey`s test p<0.01; Table S2).



Phosphate in LD A showed a higher concentration during the first hour of incubation, followed by a sharp decrease (from 0.17 to 0.05, approximately; Fig. 3a) in the treatment with copepods as well as control supported by $F_{1,20}$: 17.7 (p<0.0001). Despite significant differences in time, treatment versus control differences were not significant ($F_{1,20}$: 4.5, p=0.06). Also, no

5 significant differences in phosphate distribution through time and between treatments were found in LD B and LD C experiments (Fig. 3c, e) LD B: $F_{1,18}$: 0.13, p=0.7; LD C: $F_{1,20}$: 0.44, p=0.51). DOP concentrations (Fig. 3b, d, f) did not vary significantly trough time in the three experiments (LD A: $F_{1,20}$: 0.3, p=0.87; LD B: $F_{1,20}$: 0.5, p=0.43; LD C: $F_{1,20}$: 0.70, p=0.60), but significant differences were estimated between the treatment with copepods and control in LD B and LD C (LD A: $F_{1,20}$: 0.43, p=0.52; LD B: $F_{1,20}$: 8.92, p=0.008; LD C: $F_{1,20}$: 7.94, p=0.01).

During the experiments, the resulting N: P ratios ($NO_{2+3}^-$:$PO_4^{3-}$) of the dissolved inorganic and organic are showed in Fig. 4. In the LD A experiment the inorganic N: P ratios (Fig. 4a) were lower than Redfield (N: P= 16) during the first hour of incubation followed by a ratio increment from 12.5 to 32.4 at the end of incubation, only in the treatment with copepods, whereas a decrease in N: P ratio was observed at the end of incubation in the controls. Meanwhile, in the same experiments

in LD A, DON: DOP ratio decreased from 25 to 16.4 at the end of the experiment (Fig. 4b). In LD B experiment (Fig. 4c), lower N: P than Redfield ratio was found through the experiment, suggesting a N deficiency relative to P, except at 1 h of incubation when a higher N: P ratio was observed (2.6 times higher than Redfield) in the treatment with copepods, linked to a nitrate increment (see Fig. 2f). The DON: DOP ratios was close to the Redfield ratio in both treatments and decreased from the first hour towards the end of the incubation (Fig. 4d). Meanwhile, in LD C the organic dissolved nutrients ratio (Fig. 4f)

indicated a substantial accumulation of N with respect to P from the first hour of the incubation in the treatment with copepods, showing an important deviation from the Redfield ration (< 16:1).

### 3.3 Changes in bacterioplankton abundance during the experiment evolution

Changes in bacterioplankton abundance are showed in the Fig. 5. Initial experimental conditions showed that bacterioplankton was more abundant in LD B, followed by LD C and LD A experiments. In LD A experiment (Fig. 5a)

25 bacterioplankton presented an increase through time towards the end of the incubation, from $100x10^3$ to $200x10^3$ cell mL$^{-1}$, showing significant statistical differences between the sampling time points (F1,20: 13.5, p<0.001) but not between treatments ($F_{1,20}$: 0.004, p=0.95). In contrast, in LD B experiment (Fig. 5b), a decrease in bacterioplankton abundance (around $300x10^3$ cell mL$^{-1}$) was observed between 0.5 to 1 h and between 1 h and 2 h of incubation, in copepod and control treatments, respectively, giving significant differences both between sampling times ($F_{1,16}$: 33.8, p<0.000) and treatments

(copepods and control, $F_{1,16}$: 7.69, p=0.014) and between time. In LD C (Fig. 5c), a strong difference (> $200x10^3$ cell mL$^{-1}$) was observed between sampling times ($F_{1,20}$: 9.93, p<0.000) and between treatments and control ($F_{1,20}$: 6.22, p=0.02).



### 3.4 Active bacterial composition and their response during the incubations

The alpha diversity derived from normalized 16S rDNA and rRNA libraries is shown in the Fig. 6. The bacterial taxonomic richness determined (OTU number) and expected on the Chao1 index was higher in DNA in situ compared to the cDNA in situ and the samples related to the experiment. In accordance, Shannon diversity index showed that the bacterial community

obtained in situ from 16S rDNA reached higher values compared with the incubated community. The alpha diversity showed slight differences between treatments with copepods in the three experiments, characterized by higher values at the end of incubation in the LD A and LD C (Fig. 6a and 6c) and at the beginning of the experiments in the treatment with copepods in LD B (Fig. 6b).

The total, in situ bacterial community composition (Fig. 7a) was characterized by a higher contribution of Alphaproteobacteria, mainly by SAR 11, dominating slightly more in LD C, followed by LD B and LD A, respectively, reaching 49.8%, 41.1% and 31.1% of libraries derived from DNA (Fig. 7b). Cyanobacteria were also present in our libraries with a higher contribution in LD C (13.5%) than in LD A and LD B. In addition, a high number of abundant phyla were observed in LD A compared to the other stations. On the other hand, the active in situ bacterial composition (Fig. 7c) 16S

rRNA was consistent with 16S rDNA, with the predominance of Alphaproteobacteria and SAR11 class followed by the same trend (Fig. 7b and 7c), with higher contribution in LD C (35.5%), LD B (29.1%) and LD A (21.7%).

In general, clustering analyses showed that the active bacterial community composition varied significantly during the incubation compared to the in situ community in the three experiments (Fig. 8). During LD A incubation, the bacteria

community composition was characterized by an increase of Gammaproteobacteria from 39% to 68% in T0 and T4 in the copepods treatments (Fig. 8a). Also, a decrease in Cyanobacteria, Chloroflexi and other low frequent taxa was found. During the experiment at LD B, smaller differences in taxa contribution were observed through the incubation. Gammaproteobacteria, increased their contribution through the incubation from 28% at T0 to 32% at T4 (Fig. 8b). In contrast to the other experiments, the incubation at LD C (Fig. 8c) presented a decrease in Gammaproteobacteria

contribution and an increase in Alphaproteobacteria. The cluster analysis (Fig. 8) associated with bacterial orders contribution changes during the experiments indicate that the variability was mainly associated to the incubation time, T0, T1 and T2 versus T3 and T4, in LD A and LD B experiments, respectively. However, in LD C experiment the bacterial orders were mainly grouped by treatments without effect of incubation time (with and without copepods; Fig. 8c).

In addition, the greatest dissimilarities, based on SIMPER analysis (Fig. 9; Table S3), between copepod treatments and control in LD A (Fig. 9a) was mainly due to an increase in Alteromonadales and Oceanospirillales in the treatment with copepods, whereas SAR11 and Rhodobacterales increase their relative abundance in controls. At LD B experiment (Fig. 9b), smaller dissimilarities between treatment and control (9.5%) were observed and mainly due to a successive decrease of



Alteromonadales contribution and increase of SAR11contribution in the treatment with copepods. In contrast in the LD C station (Fig. 9c), the dissimilarities were mainly due to a longer contribution in Alteromonadales, replaced by SAR 11 at the end of the incubation in the treatment with copepods. The specific order contribution for the three experiments are shown in Fig. S2, S3 and S4, as abundant (>0.5%), semirare (0.5%-0.1%) and rare (<0.1%; Pedrós-Alió, 2012).

PCO analysis with Orders contribution and environmental parameters indicates that in LD A experiment the initial sampling points (T0, T1 and T2) were associated with inorganic compounds ($NH_4^+$, $NO_3^-$, $NO_2^-$ and $PO_4^{3-}$; Fig. 10a) for the treatment with copepods and control. Meanwhile, nitrite concentrations were associated with the latter time-points in LD B (Fig. 10b). In LD C experiment (Fig. 10c), no visual association was observed between environmental parameters and the order

contribution. In addition, the PERMANOVA main test revealed significant differences among all samples regarding the different experiments (pseudo-F=12.7, p=0.001). However, differences between treatments with copepods and control, and between sampling points did not showed significant differences (PERMANOVA, pseudo-F=2.1-1.3, p=0.12-0.25).

**4 Discussion**

The two oceanographic regions in which our experiments were performed presented contrasting trophic conditions. The MA

region being was characterized by higher nutrient concentrations particularly for nitrate in LD A, and by the presence of *Trichodesmium* spp bloom, with higher $N_2$ fixation rates (Bonnet et al., 2017; Caffin et al., this issue). In contrast, the SG region is characterized by lower nutrient concentrations compared to MA region, also, resulting in lowest N: P ratio. These conditions were reflected in the abundance and composition of zooplankton, and also in the composition of bacterioplankton. In the case of zooplankton, the average abundance in the MA region was 1,212 ind. $m^{-3}$ (sd=334 ind. $m^{-3}$, highest value 2,017

ind. $m^{-3}$), whereas in the SG region the average abundance was 665 ind. $m^{-3}$ (sd=213 ind. $m^{-3}$, highest value 409 ind. $m^{-3}$; Carlotti et al., this issue). On the other hand, the bacterioplankton community composition revealed a higher diversity in the MA region compared to SG, with a higher dominance of Alphaproteobacteria in both region, however, SAR 11 increased their contribution in the ultra-oligotrophic area (SG).

**4.1 Biogeochemical changes in response to the dissolved compounds released by copepods**

During our study, we evidenced that copepods exert an influence on biogeochemical variability during the incubations at contrasting trophic conditions including ultra-oligotrophic (SG) versus oligotrophic conditions (MA), showing significant increase of ammonium and DON accumulations. In addition, also copepods influence DON recycling but differentially at the different stations. In the experiment with copepods carried out in the LD A station, higher DON concentrations were observed at initial times (between 0 and 1 h). This DON was actively consumed between 1 and 2 hours of incubation.

However, this higher DON consumption (3.62 µmol $L^{-1}$ $h^{-1}$; Table S4) was not regenerated, since ammonium did not accumulate with time. In addition, at the same time an increment in bacterioplankton abundance was observed (22,8x$10^3$ cell $mL^{-1}$), suggesting that DON was assimilated by growing of the heterotrophic bacteria. It has been reported that zooplankton




can release highly labile compounds through their metabolism and these compounds can be rapidly assimilated by microbial communities (Arístegui et al., 2014; Vargas et al., 2007).

On the other hand, the second experiment carried out in the region MA (at LD B) evidenced a potential nitrification

associated with copepods excretion. This because, at the initial time, a decrease in ammonium concentration (between 0.5 and 1 h of incubation) related with nitrate accumulation (highly variable between the triplicates) was observed in the treatment with copepods, but not observed in the control. It has been reported that nitrification is active in upper layers in the South Pacific Gyre and often balances the biological demand for nitrate (Raimbault and Garcia, 2008). Furthermore, in eutrophic water a tight coupling between excretion and the functional microbial groups involved in ammonia oxidation has

been reported (Valdes et al., 2017a). Besides, ammonification from DON was evident at the end of incubation (1 and 2 h) at a rate of 0.01 $\mu$mol L$^{-1}$ h$^{-1}$ (Table S4). As in LD A, this demand could be probably associated with heterotrophic bacterioplankton, but in considerably lower rates in this experiment than in LD A.

The SG experiment with copepods (at LD C station) presented the highest evidence of DON

remineralization/ammonification based on the differences in ammonium concentrations between copepods treatments and control up to 0.3 $\mu$mol L$^{-1}$, several times higher than in situ concentration (Table 1). DON consumption was observed through the first 2 h of incubation in the treatment with copepods (Fig. 2; Table S4). In addition, as in LD B, nitrate accumulation was observed at the end of incubation in the treatment with copepods compared to the control. Our results suggest presence of two processes which could occur in the ultra-oligotrophic station in the SG: the release of DON by

copepods stimulating ammonium regeneration by bacteria, and the increment in nitrate concentration strongly linked to the nitrification process. Thus, DON remineralization could provide additional substrate for microbial and phytoplankton growth in LD C station.

Thus, DON excretion by copepods in oligotrophic and ultra-oligotrophic water could play a major role in nitrogen cycling.

In addition, the highest differences in DON concentration between the treatment with copepods and the control (without copepods) were found at initial time points, mainly in LD A and LD C experiments. Zooplankton can excrete DON in highly concentrated spurts lasting up to an hour, unlike ammonium which is excreted continuously by crustacean (Gardner and Paffenhöfer, 1982; Steinberg et al., 2002). In this study, we cannot attribute the increase in the concentration of the different compounds only to the excretion process since we did not monitor possible leaching from fecal pellets produced during the

experiments. The leaching is also one of the possible pathways through which zooplankton can release dissolved compounds, along with the sloppy feeding and excretion. However, it has been demonstrated that the contribution provided by leaching is insignificant compared to excretion, less than 6% in the case of the release of DON, whereas ammonium was undetected (Saba et al., 2011; Steinberg et al., 2000). Thus, if we consider leaching from fecal pellets as negligible, then DON accumulation could account for up to 1.91 $\mu$mol L$^{-1}$ h$^{-1}$ excretion in the LD A experiment. However, the experimental



copepod densities were several orders of magnitude higher than those expected in situ (10 copepods for 0.5 L volume, i.e. 20,000 ind. m$^{-3}$), so that our results might be overestimated because copepods concentration in our bottles did not reflect the in situ abundance (MA: 1,500 ind. m$^{-3}$ and SG: 500 ind. m$^{-3}$; Carlotti et al., this issue). However, such concentration was strictly necessary through the experiments to obtain a signal in ammonium concentrations, which were very low in the study

area (Table 1). In addition, we cannot estimate excretion rates for all compounds since they appear later in the incubation probably modulated by microbial metabolism.

On the other hand, DOP showed significantly higher concentrations in the treatment with copepods compared to the control in the three experiments, although not over the whole incubation period but over shorter durations. In LD A experiments,

phosphate was found in higher concentrations in the treatment with copepods compared to the control between 0 and 1 h of incubation. In LD B and LD C experiments, concentrations in the treatment with copepods versus control were significantly higher at the beginning and at the end of the incubation for LD B, also at the end of incubation for LD C. Our results suggest that copepods could potentially contribute with substantial amount of DOP and provide an alternative source of phosphorus for phytoplankton and microbial loop in this study area. During LD B station, a bloom of *Trichodesmium* spp was observed,

and the satellite data suggested that the bloom was already two months old by the sampling time at station LD B (de Verneuil et al., this issue), which is in agreement with our lower nutrients concentrations obtained in situ and through our incubation in this station. DOP concentration was significantly higher in the treatment with copepods than in the control in our incubation at this station. The contribution of DOP by copepods could contribute to maintain the longer bloom in this region (MA), providing an additional source of phosphorus for biomass growth. It has been reported that *Trichodesmium* spp

are capable to hydrolyze DOP compounds (Mulholland et al., 2002), thus zooplankton metabolism could potentially supply a significant portion of the cellular phosphorus necessary for their growth.

**4.2 Changes in bacterial community composition induced by dissolved compounds released by copepods**

In our experiments, the bacterioplankton abundance followed the same patterns of variation in the experiments with copepods and in the control, for each of the three experiments, even if the abundance was slightly increased in the

experiments with copepods. In our LD B and LD C experiments, this impact on bacterial community structure was significantly different with additions of copepods compared to the control.

Through the incubations, the bacterioplankton community was characterized by changes mainly in Gammaproteobacteria and Alphaproteobacteria in all stations. An increase of Gammaproteobacteria is observed at the beginning of LD A

experiment, mainly associated with the Alteromonadales and Oceanospirillales orders, which increased their contribution in the treatment with copepods compared to the control in 42 % and 7%, respectively (Fig. 9). The faster response in Alteromonadales (Gammaproteobacteria) has been documented and are frequently observed in microcosm studies, due to their high growth rate and their ability to exploit DOM rapidly when is available (Alonso-Saez and Gasol, 2007; Fuchs et al.,



2000; Landa et al., 2013). The high DON demand found in this incubation (LD A) could be related to an increase in Alteromonadales.

In contrast, in the experiment carried out in LD B station, a substantial decrease in Alteromonadales and Oceanospirillales

was observed in the treatment with copepods compared to the control. The higher contribution through the time was due Alphaproteobacteria, in which SAR11 increase their relative abundance (~8%) at 1 hour of incubation and Caulobacterales (<0.1%) at the end of incubation. SAR 11 was one of the major constituent of 16 rRNA (in situ and microcosm incubation) and 16 rDNA in situ and it has been reported that SAR11 is the most abundant bacterioplankton in the ocean, is well adapted to lower concentration of nutrients, and are specialized to oxidize many labile and low-molecular-weight compounds

produced by other plankton (Giovannoni, 2017). In addition, during our incubations, SAR11 increased their contribution only in LD B experiment which coincided with the station with lowest nutrients concentrations. Also in this experiment, the bacterioplankton abundance showed a strong decrease in both treatment and control at 1 hour of incubation.

On the other hand, in the LD C experiment, an increase in Alteromonadales and Vibrionales were observed.

Alteromonadales decrease their contribution in the treatment with copepods at the end of incubation, meanwhile increase their contribution in control. The reverse pattern was observed for SAR11 which increase their contribution at the end of incubation. Based in our experiments, Alteromonadales increase is in coincidence with DON available from copepods excretion during the first 2 h of incubation, afterwards at the end of the experiment nutrients were diminish, conditions that favored the versatile SAR 11. Peduzzi and Herndl (1992) observed high monomeric carbohydrate concentration and bacterial

activity in experiments where copepods were included. Furthermore, these authors observed that bacterial communities living in oligotrophic areas can be efficient to utilize the newly available substrate source, according with our results. Our results suggest that copepods can provide substantial amounts of nitrogen and phosphorus ($NH_4^+$, DON, DOP) which microbial communities can directly use in a short period of time enhancing the bacterioplankton remineralization in the WTSP.

Bacterial remineralization of DOM derived from copepods metabolism could be a highly efficient mechanism to maintain the nutrients in the upper layer supporting the phytoplankton and microbial growth. In this sense, the response associated with the different regions of our study suggests that copepods in the MA region could provide substantial amounts of DOP as an alternative substrate for phytoplankton and bacterial growth, while the enhanced remineralization by copepods could be

more important in the ultra-oligotrophic area of the SG region. Furthermore, the biogeochemical impact of copepods metabolism may not be limited to the upper layer, as zooplankton can move through the water column by diel vertical migration, promoting the export of dissolved compounds through their metabolism at deeper layers. We conclude that zooplankton metabolism seems to play a critical role in nutrient recycling by fuelling microbial and phytoplankton growth in the vast oceanic regions of the world ocean.




*Acknowledgement*. This is a contribution of the OUTPACE (Oligotrophy from Ultra-oligoTrophyPACific Experiment) project funded by the French research national agency (ANR-14-CE01-0007-01), the LEFE-CyBER program (CNRS-INSU), the GOPS program (IRD) and the CNES (BC T23, ZBC 4500048836) and by European FEDER Fund under project

1166-39417. The OUTPACE cruise was managed by T. Moutin and S. Bonnet from the MIO (Mediterranean Institute of Oceanography). The authors thank to the crew of the R/V L´Atalante for outstanding shipboard operation. Dr. Mireille Pujo-Pay (Observatoire Océanologique de Banyuls sur mer), Olivier Grosso (MIO) and Sandra Helias (MIO) are thanked for their assistance during the analysis of nutrients. This work was supported by the Comisión Nacional de Investigación Científicas y Tecnológicas (CONICYT) through Grants FONDECYT No 1130511 and 1150891 and Instituto Milenio de Oceanografía

(IMO) Grant IC120019. Additional support has been provided by FONDAP grants IDEAL and INCAR (15110027 and 15150003, respectively), and CONICYT PIA PFB-31 COPAS Sur austral program. The work was developed in the frame of LIA MORFUN. Valdés and Donoso were funded by CONICYT Scholarship

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



**Table 1:** Deep chlorophyll-a maximum (DCM) depth, ammonium ($NH_4^+$), nitrate ($NO_3^-$), nitrite ($NO_2^-$), phosphate ($PO_4^{3-}$) concentration and N: P ratio at each sampling site.

|  | LD A | LD B | LD C |
|---|---|---|---|
| DCM (m) | 80 | 34 | 140 |
| $NH_4^+$ [$\mu$mol L$^{-1}$] | 0.025 ± 0.001 | 0.011 ± 0.00 | 0.017 ± 0.000 |
| $NO_2^-$ [$\mu$mol L$^{-1}$] | 0.064 ± 0.002 | 0.023 ± 0.02 | 0.048 ± 0.004 |
| $NO_3^-$ [$\mu$mol L$^{-1}$] | 1.53 ± 0.008 | 0.051 ± 0.021 | 0.247 ± 0.012 |
| $PO_4^{3-}$ [$\mu$mol L$^{-1}$] | 0.16 ± 0.012 | 0.03 ± 0.025 | 0.243 ± 0.008 |
| N: P | 9.97 ± 0.70 | 4.52 ± 4.07 | 1.21 ± 0.11 |

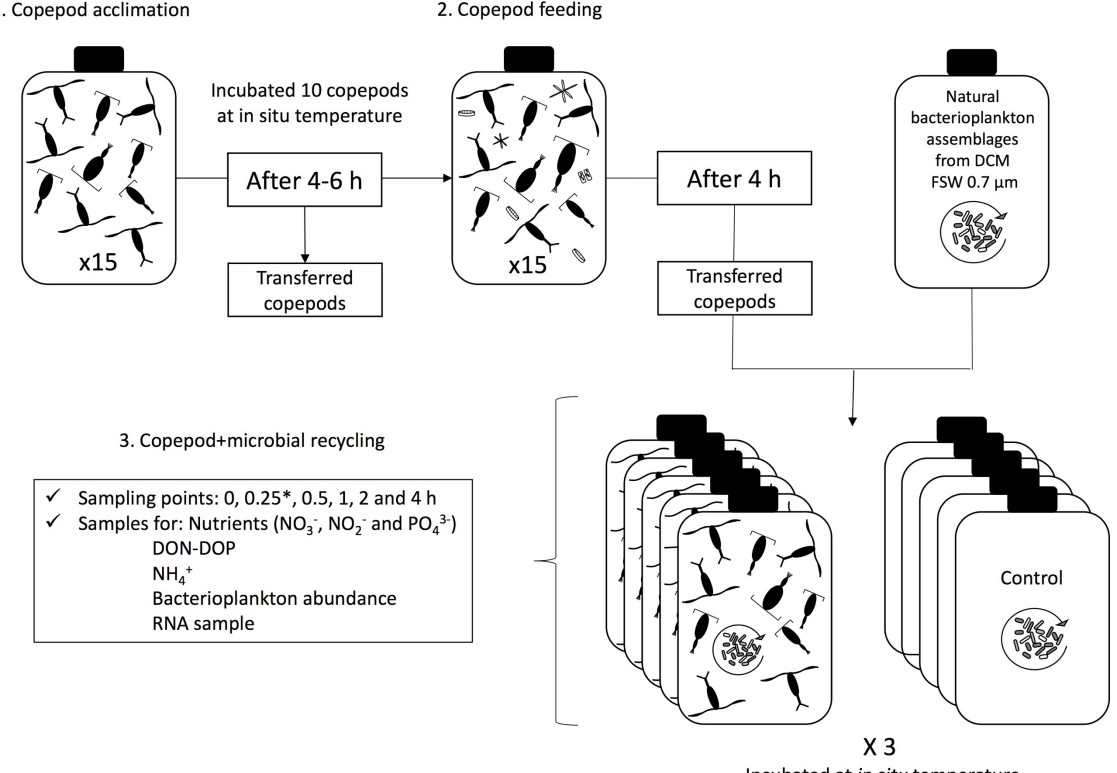

**Figure 1:** Experimental design. The design consisted of three phases done sequentially. Each bottle with copepods was acclimated for at least 6 hours (phase 1), followed by feeding (phase 2) with the Deep Chlorophyll-a Maximum (DCM) water for 3 hours. Copepod+microbial recycling (phase 3) was studied by adding natural bacterial assemblages from DCM filtered seawater (DCM FSW) to the 15 bottles with fed copepod, and maintaining bottles with natural bacterial assemblages and without copepods (controls). Three bottles with copepods and three control bottles were sacrificed at each of the 5-sampling time. An additional time (0.25 h) was added for ammonium determination in LD B and LD C.





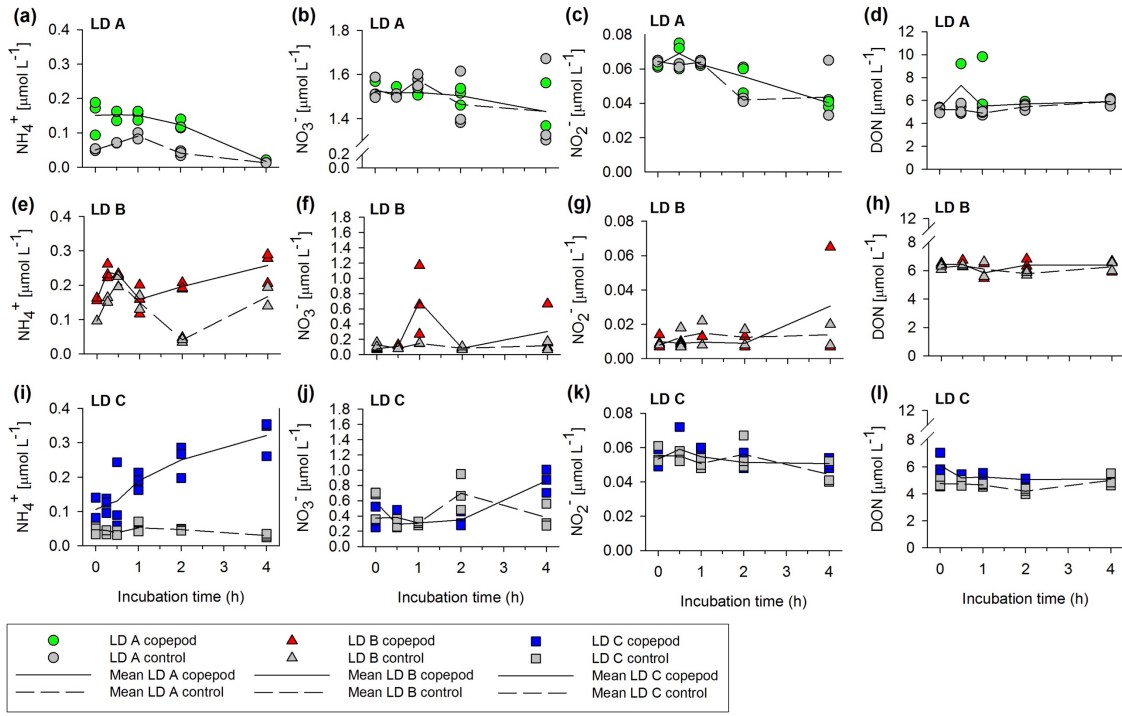

**Figure 2:** (**a, e, i**) Ammonium (NH$_4^+$), (**b, f, j**) nitrate (NO$_3^-$), (**c, g, k**) nitrite (NO$_2^-$) and (**d, h, l**) dissolved organic nitrogen (DON) variability through the incubation for treatments with copepods and control (without copepods) at each LD experiment.



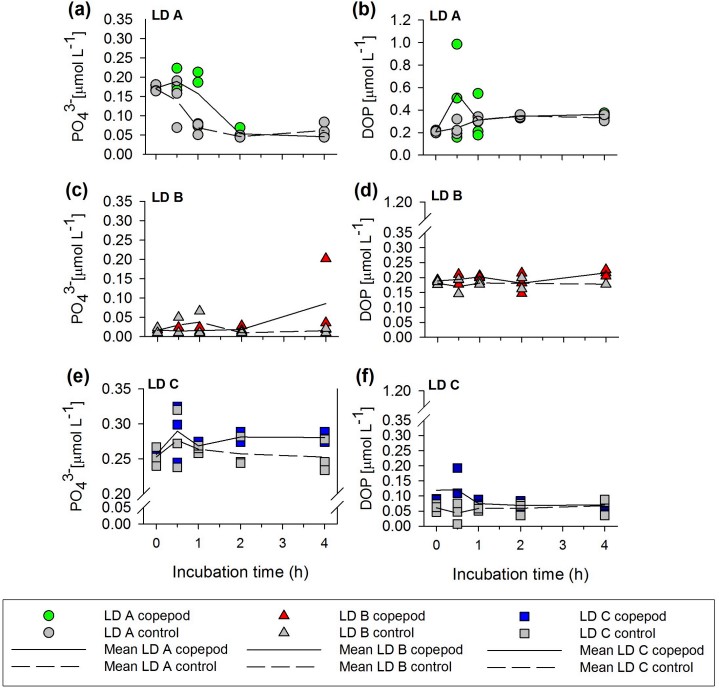

**Figure 3:** (**a, c, e**) Phosphate ($PO_4^{3-}$) and (**b, d, f**) dissolved organic phosphorus (DOP) variability through the incubation for treatments with copepods and control (without copepods) at each LD experiment.





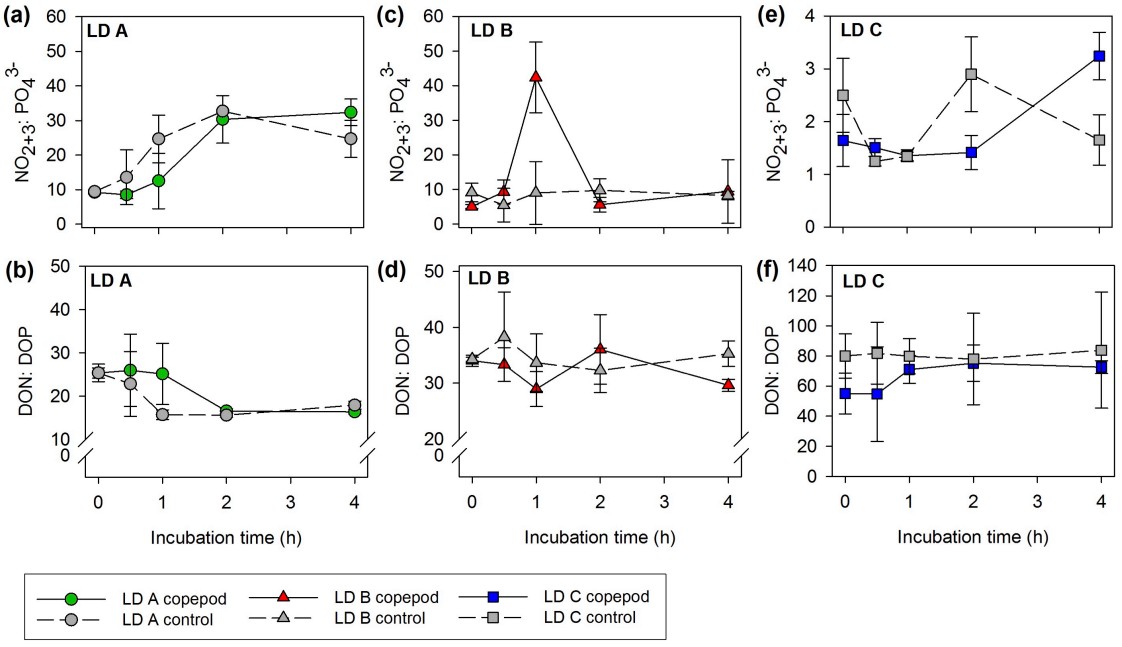

**Figure 4:** (**a, c, e**) N: P ratios of the inorganic (NO$_{2+3}^{-}$: PO$_4^{3-}$) and (**b, d, f**) organic (DON: DOP) nutrients through the incubation for treatments with copepods and control (without copepods) at each LD experiment.

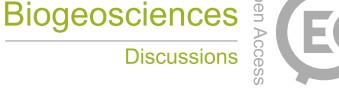



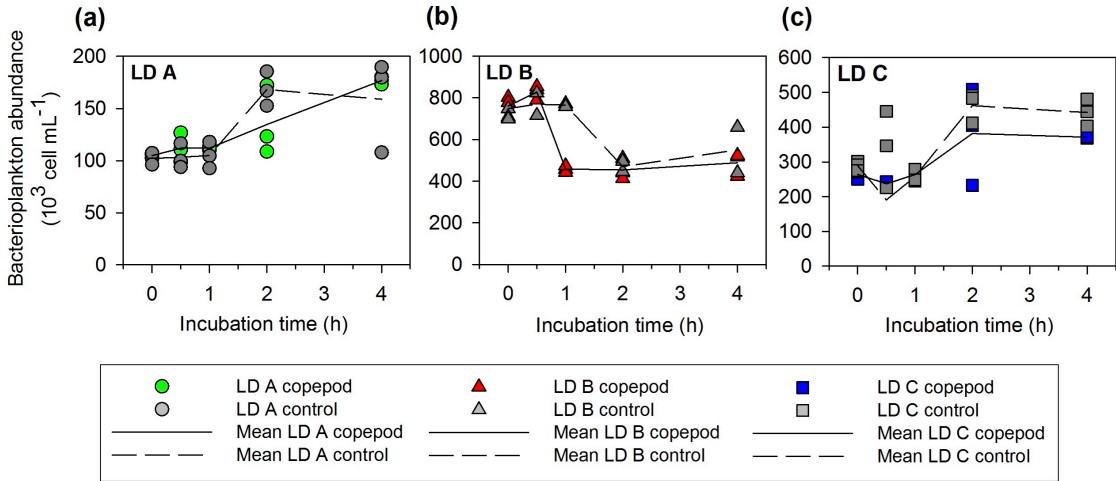

**Figure 5:** Bacterioplankton abundance through the incubation for treatments with copepods and control (without copepods) at each LD experiment. (**a**) LD A, (**b**) LD B and (**c**) LD C.





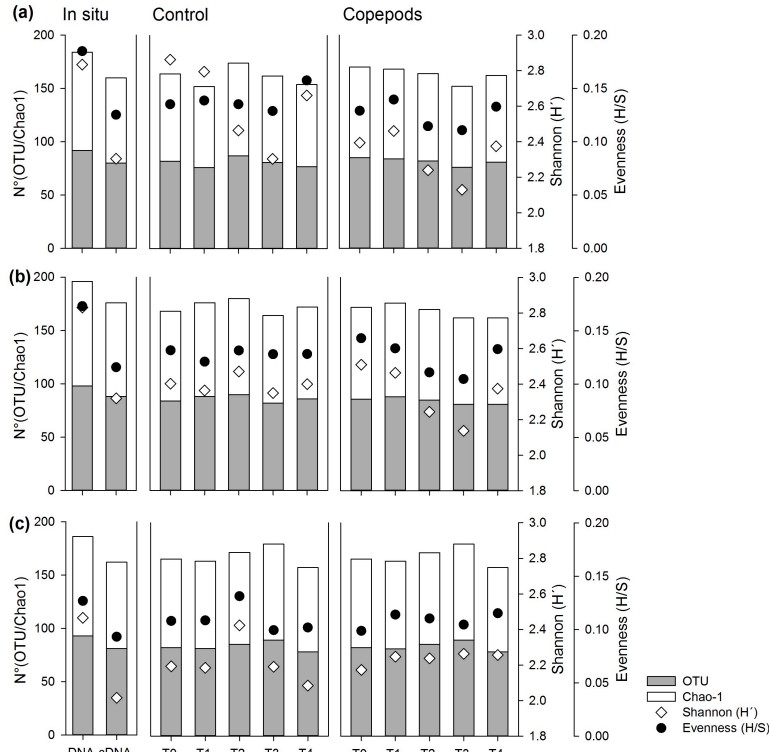

**Figure 6:** Richness, Chao1, Diversity (Shannon H´) and Evenness from the total and active in situ bacterial community and through the incubation for the active bacterial community at each LD experiment: (**a**) LD A, (**b**) LD B and (**c**) LD C.



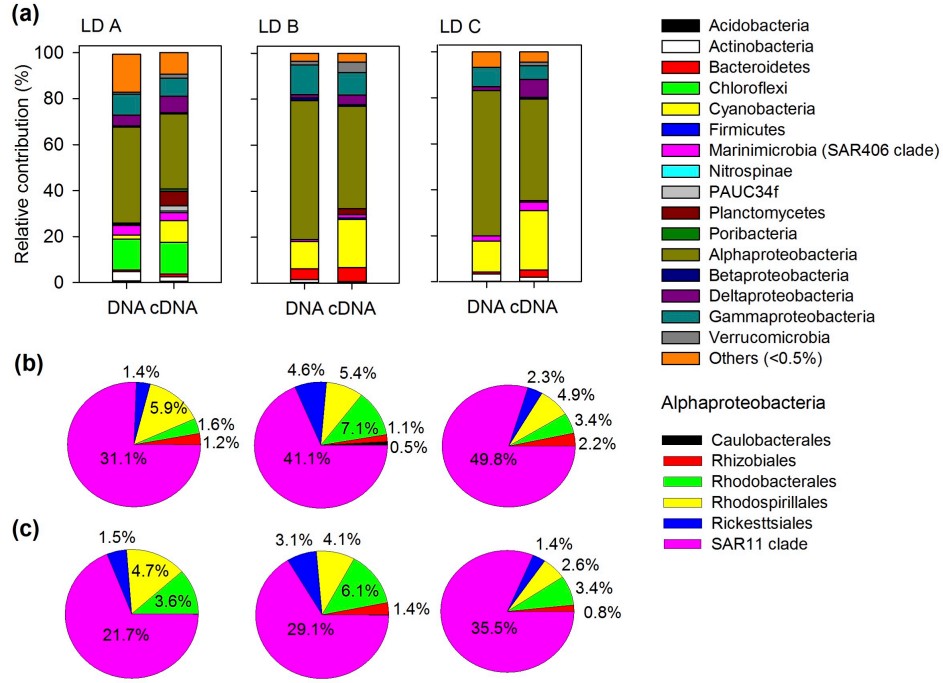

**Figure 7:** (**a**) Relative abundance of major bacteria phyla and Proteobacteria class at each LD station, with a zoom in Alphaproteobacteria order at initial in situ conditions for the three-sampling site: (**b**) total Alphaproteobacteria (DNA) and (**c**) active Alphaproteobacteria community (cDNA).





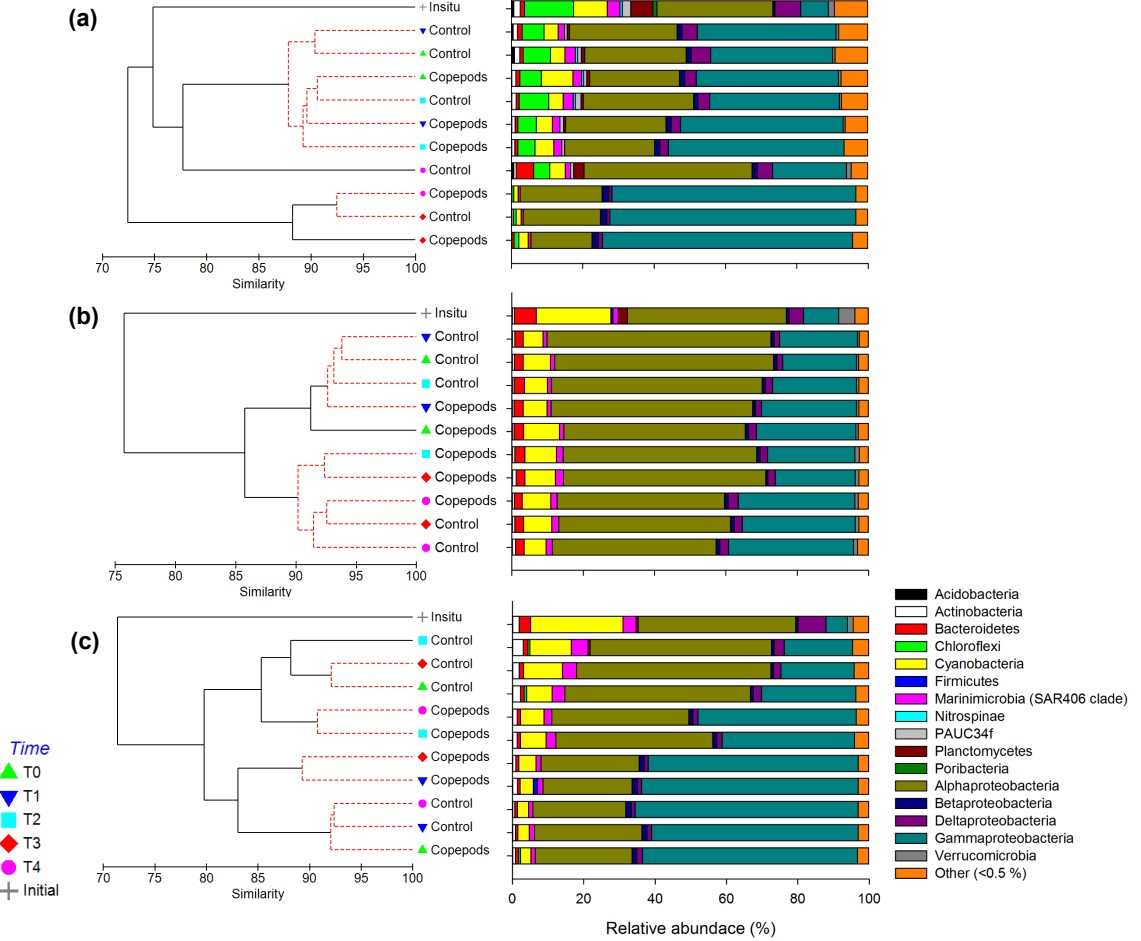

**Figure 8:** Similarity dendrograms based on the 16S rRNA and relative contribution of the abundant bacteria phyla and Proteobacteria subclasses for the initial in situ community, treatments with copepods and controls (left panel) through the incubation for the three experiments (**a**) LD A, (**b**) LD B and (**c**) LD C. Clustering in one the basis of a distance matrix computed using the Bray-Curtis index of similarity. The dendrogram was inferred with the unweighted pair-group average algorithm (UPGMA). Bacterial communities in the samples connected with red branch lines are not significantly different (SIMPROF test, p<0.05).





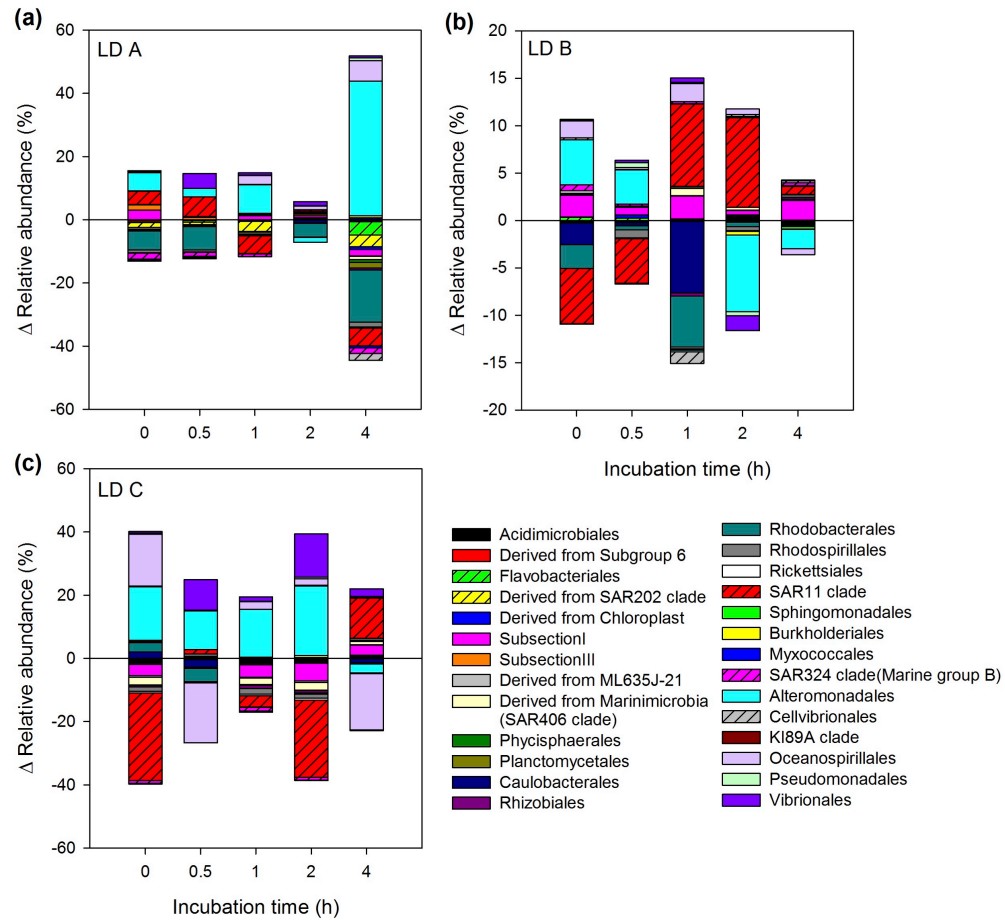

**Figure 9:** Contribution of active Orders through the incubation based on the SIMPER results. List of the OTUs explaining 50% of the dissimilarity observed through the experiment between treatment with copepods (top) and control (bottom) for each LD experiment: **(a)** LD A, **(b)** LD B and **(c)** LD C.





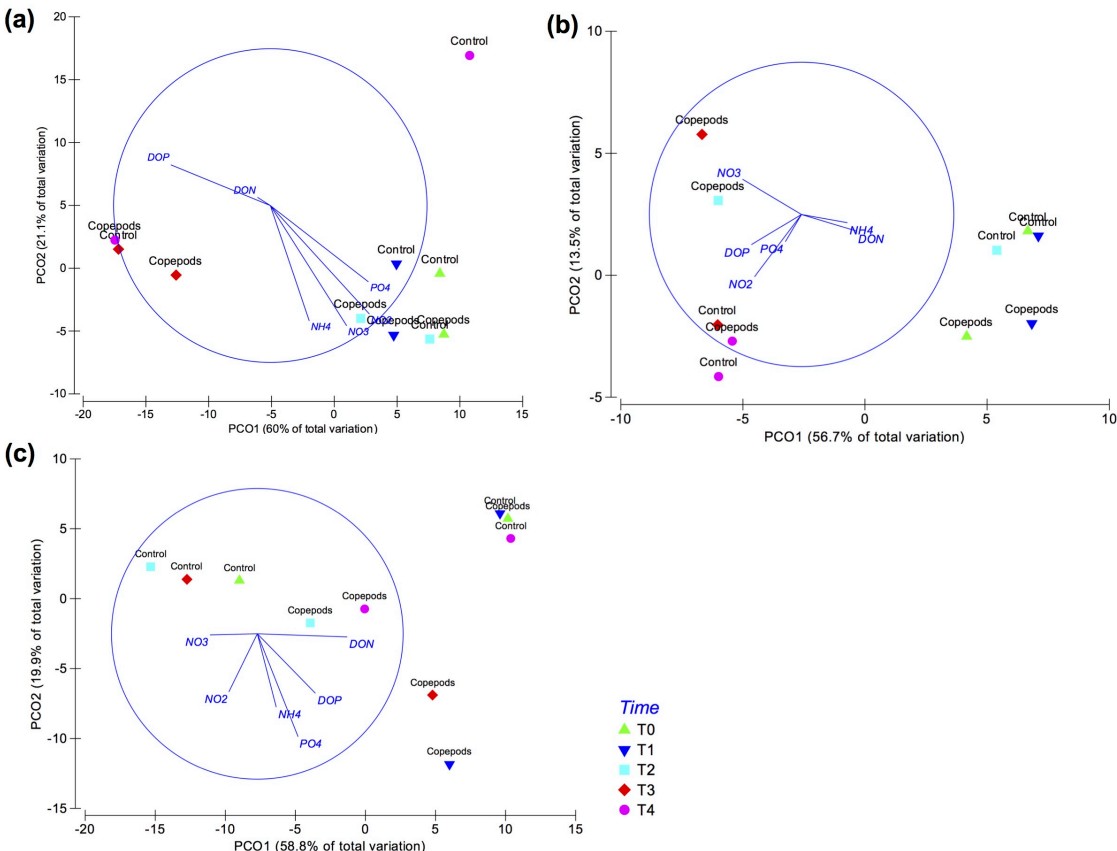

**Figure 10:** Principal component ordination based on Bray-Curtis similarity at order taxonomic level, of treatment with copepods and control (**a**) LD A, (**b**) LD B and (**c**) LD C. Vectors indicate the best environmental variables (normalized transformed) correlated with ordinations and vector lengths correspond with the correlation values.