# Peer review of "Nitrogen and phosphorus recycling mediated by copepods and response of bacterioplankton community from three contrasting areas in the Western Tropical South Pacific (20°S)"

_Biogeosciences, 2017_

## Referee Comment (RC1) · Anonymous Referee #1 · 28 Feb 2018

In the role of zooplankton in nutrient regeneration, metabolic studies are always important and useful, especially when they concern areas with relatively few published data, measurements of multiple parameters, and effects on the microbial community. Generally, there are some nice parts in the manuscript, drawing together from diverse sets of original information. However, there are important issues to be dealt with, prior to publication.

1) MAJOR COMMENTS

My main concern is the use of data from only 3 time-space points (3 stations in spring 2015 and water collected only from DCM) to make conclusions on the importance of

copepods in N and P recycling for the whole Western Tropical South Pacific (title, p. 13 lines 22 to 24). These 3 stations are located (stations coordinates not given) on a transect > 1300NM long, thus covering a very vast area. To support the potential in situ applicability of their conclusions, the authors should give strong evidence (based on other manuscripts of the same issue or at least based on previous bibliography) about the temporal and spatial (horizontal and vertical) homogeneity of this vast area.

My second concern, which enhances the first one, is using an experimental ratio of copepods to bacteria > 13:1 compared to the one in the field (p.12 lines 2 & 3), to make conclusions on the in situ effect of zooplankton on the microbial community. It is fully understandable that high experimental densities of copepods are necessary to obtain a signal in nutrients within a few hours. However, it is doubtful that if the nutrients available to bacteria per time unit were much less, that the remineralisation effects and shifts in bacterial composition would be the same (especially considering also the role of the other player - phytoplankton).

In brief, the transfer of lab observations under such experimental conditions and from only 3 space-time points to processes occurring in the field at a very different copepod : bacteria ratio and over vast spatio-temporal scales is not at all straightforward. My suggestion is to rewrite conclusions in a much more conservative way. Then the manuscript including title should be adapted in consequence.

2) SPECIFIC COMMENTS

2a) Introduction :

p.2 lines 20, 25, 26. Copepods may excrete much more than 53% of their body nitrogen in the form of ammonia and this percentage is highly variable (as said in line 25). There are many papers on this subject, including review papers.

p.3 lines 16-17: How is the presence of an oligotrophic gradient supported? Nutrient values from the only three stations are insufficient. Nutrient and/or Chla data from more

stations would be helpful.

2b) Methodology:

General remark for the experimental set up: an additional control with copepods only would have being very helpful.

p.3 line 22 : please define "long duration"

p.3 line 22 : please add coordinates of the stations

p.3 lines 24-25 : please add maximum and minimum values of chlorophyll-a

p.4 lines 2-4 : please specify if tow was vertical (or oblique), tow speed, net diameter

p.4 line 13 : the field composition of copepods and other zooplankton should be also shown in Table S1

p.4 lines 23-24 : please specify % of mortality

p.8 lines 21 : check that Redfield ratio of organic nutrients is < 16:1

p.9 lines 7-8 : unclear meaning

2c) Results:

General remark. Do not repeat values that can be found in tables or figures unless necessary.

Table 1 : use either 2 or 3 decimals depending on the precision of the method for each parameter

Table 1 : add Temperature, Chla, DON, DOP, values at DCM

p.7 line 25 : "first treatment", replace by "treatment with copepods"

p.8 line 4 : "significant difference in time,". Please add results of statistical test

2d) Discussion:

General remark. Do not repeat detailed description of results, but only briefly giving outcome in connection with related literature.

p.10 line 15 and line 17: Table 1 shows that MA stations (LD A & LD B) are characterized by higher inorganic nitrogen but not higher inorganic phosphorus concentration than SG (LD C).

p.10 line 25: "influence on biogeochemical variability": not clear.

p.10 lines 31-32: Since there was no significant difference between treatments in LD A for bacterioplankton abundance (p.8 lines 26-27), a conclusion on an effect of zooplankton cannot be made.

p.11 lines 7 to 10. LD B is not in the South Pacific Gyre (p.1 line 24). Check and eventually move this part to discussion on LD C which is in the South Pacific Gyre.

p.13 lines 33-34: Too general conclusion on the role of zooplankton metabolism, not supported by manuscript's observations.

2e) Bibliography:

References to articles in preparation should be avoided

3 ) TECHNICAL CORRECTIONS

p.2 line 2, p.8 line 18, p.10 line 12, p.10 line 15, p.12 line 32, p.13 lines 8-9: English grammar mistakes

p.2 line 10 : change order of references

p.3 line 3 : "...in a phosphorus limited..."

p.3 line 5 : "...Gasol, 2007"

p.3 line 8 : "...zooplankton-produced DOM by bacterial communities..." : unclear expression

p.4 line 14 : "acclimation" instead of "acclimatization"

p.4 line 21 : "such that 6 bottles were added in that case" : not clear

p.5 line 23 : correct "u sing"

Please correct typing mistakes and journal names abbreviations in bibliography

---

## Referee Comment (RC2) · Anonymous Referee #2 · 2 Apr 2018

The authors present a study that simultaneously examines the role of zooplankton and bacteria on nitrogen and phosphorus cycling in low-nutrient systems. Few studies have tried to assess these factors simultaneously to link the two processes. Although the authors provide a lot of interesting results, I am not convinced that their data back up the larger assertions of the role of copepod excretory products play in spurring growth of microbes. Also, I am not convinced that in situ conditions were accurately enough recreated to directly apply these results to the in situ setting.

More specifically:

1) It is challenging to know how much N and P is being utilized by the bacteria (and in

what forms) when you don't have baseline data for N and P excretion rates for copepods in the absence of microbes. Without this information, you are unable to say the actual impact either party (copepods or microbes) is having on buildup or drawdown of different N or P compounds.

2) In many cases, there were not significant or clear differences between the controls and copepod treatments. For instance, the bacterial community structure for LD A and LD B stations grouped more by time point than by treatment. That implies an effect of the incubation itself, and not due to extra nutrients brought on zooplankton excretion. Also, in 2 out of the 3 experiments, it appears that bacterial abundance was higher in the controls at the end of the experiment. If they were using the nutrients to grow, we would expect to see significantly higher abundances in the copepod bottles.

3) Copepod densities in the bottles were much higher than in situ conditions. While high densities of zooplankton are definitely necessary for detectable changes in excretory products, this also meant that the change in N and P facilitated by zooplankton is likely to be much larger within this experiment than what bacteria experience in the field.

4) When looking at the dendrogram results in Figure 8, it appears that the DNA samples taken from the in situ environment show a very different community structure when compared to the communities that were present in the incubation bottles at time T0. If the community structure in the bottles at the beginning of the experiment isn't representative of the local community, the changes observed might not reflect what would occur in situ.

5) I am concerned that there are significant differences between copepod and control treatments at T0. The methods say that the organisms were placed in new incubation water at the beginning of the excretion step. I assume initial samples were taken immediately. While the authors attempt to explain the differences seen in DON (p. 11, line 26), there is no information addressing the differences in bacterial community structure at T0 for several of the experiments.

Additionally, the entire document needs to be reviewed for use of proper English grammar. Most of the issues are minor, but do make the paper a bit harder to read and understand. I will point out several below in my comments, but there are certainly more within the document.

Specific Comments:

Introduction:

*P. 2, line 7-8: Citation needed for sentence about thermal stratification and global warming decreasing the nutrient supply

*P. 2, lines 19-23. There has been a lot of work on zooplankton excretion and the various products. A more comprehensive summary should be provided about N cycling of body nitrogen and support for phytoplankton needs in oceanic environments.

*P. 3, line 1: "enhance" should be "enhances."

Methods:

*P. 3. Second sentence of section 2.1: Should read "The transect began west of New Caledonia and ended near Tahiti."

*P. 3, section 2.1: You should include latitude/longitudes for your LD stations. Also, you should include a map of the transect and your specific sites used for the LD experiments.

*P. 3, line 26: it says that you used a CTD rosette to collect water. Further information on hydrographic parameters should be included. Especially on chlorophyll concentrations within the DCM.

*P. 4, line 7: was the incubation seawater also collected from the DCM? I assume so, but that is not clearly stated.

*P. 4, line 8: isn't the acclimatizing step number 1?

*P. 4, line 8-9: If you were using the filtered seawater, it should read "seawater (22.5 L) was immediately filtered through a 0.7 um filter…."

*P. 4, line 10: 25C doesn't really seem cold. Perhaps "temperature controlled room" is a better term than "cold room?"

*P. 4, line 18: by controlled temperature, is that also in the 25 degrees C room? You should specifically say the temperature.

*P. 4, line 21: what 6 bottles were added? Were they all controls? You need to explain this further—it is confusing as currently written.

*P. 4, line 24: should be "to ensure" instead of "of ensuring"

*P. 4, line 26: suggest specifying "During the excretion phase…"

*P. 5, line 12: I am not familiar with the wet oxidation method, so I looked up the paper you cited. It looks that paper assesses PON and POP. What modifications need to be made to look at dissolved materials?

*Sections 2.5-2.6: My expertise is in zooplankton physiology, so I cannot assess the suitability of the genetics and bacterial methods.

*P. 5, line 22: "u sing" should be "using"

Results:

*Generally speaking, the authors did a good job displaying a very complicated data set. However, there are some places where the clarity of the results presentation can be improved with some minor modifications. I provide some of those suggestions below for Table S2 and Figures 2-5 and 8.

*P. 7, line 21: If these are ANOVA results, you should say so.

*P. 7, line 24: "notorious" is not a correct term to use. "Notable" instead?

*P. 7, line 30: I don't understand why there is a difference at time T0. Is there a gap

between when you started the experiment and took your initial sample? If that is the case, you need to specifically mention that in the methods.

*P.7 , line 30-31: Your final sentence appears to contradict itself. If there were non-significant differences, wouldn't the p-value need to be >0.05?

*P. 8, line 1-2: Consider re-wording for clarity. Specifically, all of the statistical test results should be together within parentheses.

*P. 8, line 6: missing a "(" before the "LD B…"

*P. 8, line 7: "trough" should be "through"

*P. 8, line 8: "estimated" does not make sense here. Perhaps "significant differences were only observed between…"

*P 8, second paragraph:

(1) You should be very specific here about which N:P ratio you are discussing at any given time. It is confusing at the beginning when you define it only using nitrate/nitrite and phosphate, but then discuss inorganic and organic ratios.

(2) Would you expect DON and DOP to be comparable to Redfield? Urea (one of the more common forms of DON), actually has 2 N per molecule, so if it is molar ratios that you are using, then it is not a 1:1 comparison that could be made between DON and DOP. Are there other papers that have using the organic N:P ratio? If so, it would be really helpful to bring those in to support your use and interpretation of this ratio.

(3) Line 18: you say that the DON:DOP is close to Redfield. But, the numbers that you mention in the prior sentences (and those in figure 4) are much higher than Redfield.

(4) Line 20-21: The deviation from Redfield appears to be that LD C is much higher, not lower.

(5) Line 21: should be "ratio" not "ration"

*P. 8, line 23: should be "shown" not "showed"

*P. 8, line 24: "plankton" is a plural term, so it should be "bacterioplankton were more abundant..."

*P. 8, line 30: is "and between time" referring to a different time than the sampling times you mention earlier in the sentence?

*P. 9 , line 14: On the other hand makes it sound like this is contradictory result. The two genetic markers seem to be showing similar trends. "Additionally" might be a better way to begin this thought.

*P. 9, line 27: if LD A and LD B show variability associated with time points, can you draw conclusions then based on presence of copepods? Or is this an artifact of the culturing conditions?

*P. 10, line 2: do you mean "larger" instead of "longer?"

*P. 10, line 12: "show" instead of "showed"

Discussion:

*Section 4.0: Can you say the MA region had higher nutrient concentrations when LD B had the lowest of all of your sites?

*P. 10, line 22: "region" should be "regions"

*P. 10, line 27: You said there was significant increase in DON. But this was only during one time point. The other one is during timepoint T0, which seem suspicious.

*P. 10, line 31: do you mean "increase" instead of "increment?"

*P. 10, line 31: is 22,8x10^3 correct? Should it be 228 or 22.8? The formatting is a bit odd and makes me think it is a typo.

*P. 10, line 32: You suggest DON assimilated by growing bacteria. But it is mostly in the control of LD A that the bacterial abundances increase between 1-2 hours.

*P. 11, line 10: But in LD B, you said no significant differences were observed over time (although your reported stats results don't match that - p 7, line 30)

*P. 11, line 26: How long after adding the copepods did you take your initial measurements? Could it really happen that fast? You need some more specific information to back up this as a reasonable assertion.

*P. 11, line 27: "crustaceans" instead of "crustacean"

*P. 12, line 10: Is this supposed to be phosphate or DOP? All the rest of the P references in this paragraph are to DOP, so I wanted to double-check.

*Section 4.2, first paragraph:

(1) But it didn't seem like bacterial abundance was increased within the copepod bottles for most of the timepoints.

(2) You said in your bacterial community results that LD B experiments grouped by time primarily, not by treatment.

*P. 13, line 18: "diminished" instead of "diminish"

*P. 13, line 28: Define "substantial amounts." You need to be able to better quantify this. I don't really think this statement is firmly backed up by the data presented

*P. 13, lines 32-34: It didn't really seem like you saw bacterial growth overall. This statement is not directly supported by your results.

Tables and Figures:

*Table 1: Redefine what this N:P ratio includes in the table caption.

*Figure 1:

(1) You mention "in situ temperature" a few times. What is that temperature?

(2) What does "*" mean after the 0.25 sampling point?

(3) In the caption, it says the feeding phase lasts for 3 hours. But other places in the document it states 4 hours.

(4) Specify in the caption that each of the 15 bottles contains 10 fed copepods.

*Table S2: You should include the overall ANOVA results as part of Table S2. It would be much easier for the reader if we could find all of those statistical results in one place. Perhaps on the gray lines?

*Figures 2-5: The data are really complicated and results are hard to grasp between the text (where ANOVA are reported), table S2 (where post-hoc data are reported), and graphs (where visual trends are reported). Can you find a way to mark statistically significant differences on these graphs? Then the reader can go to these for a summary.

*Figure 8: Can you use the prefixes in the descriptions of each line? "T1 copepod" or "T2 Control." It makes it very difficult to see the trends you mentioned when you have to look back at the legend each time you read a line to figure out the color-coded timepoint.

---

## Author Response (AR1)

RC1

MAJOR COMMENTS

My main concern is the use of data from only 3 time-space points (3 stations in spring 2015 and water collected only from DCM) to make conclusions on the importance of copepods in N and P recycling for the whole Western Tropical South Pacific (title, p. 13 lines 22 to 24). These 3 stations are located (stations coordinate not given) on a transect > 1300NM long, thus covering a very vast area. To support the potential in situ applicability of their conclusions, the authors should give strong evidence (based on other manuscripts of the same issue or at least based on previous bibliography) about the temporal and spatial (horizontal and vertical) homogeneity of this vast area.

My second concern, which enhances the first one, is using an experimental ratio of copepods to bacteria > 13:1 compared to the one in the field (p.12 lines 2 & 3), to make conclusions on the in situ effect of zooplankton on the microbial community. It is fully understandable that high experimental densities of copepods are necessary to obtain a signal in nutrients within a few hours. However, it is doubtful that if the nutrients available to bacteria per time unit were much less, that the remineralisation effects and shifts in bacterial composition would be the same (especially considering also the role of the other player - phytoplankton).

In brief, the transfer of lab observations under such experimental conditions and from only 3 space-time points to processes occurring in the field at a very different copepod:bacteria ratio and over vast spatio-temporal scales is not at all straightforward. My suggestion is to rewrite conclusions in a much more conservative way. Then the manuscript including title should be adapted in consequence.

Attending to the general comments:

First, the title has been modified to better fit to the paper content which mostly focuses on comparisons of remineralization and bacterial response in three contrasted areas.

The conclusion has been rewritten and now focused on the observations made at each station, avoiding general conclusions for the whole study area. These stations were selected for their different biogeochemical characteristics and chosen along an oligotrophic gradient, so that our study was performed over a heterogeneous and contrasting region. Studies in this region are very scarce and even scarcer for the role of zooplankton in nutrient recycling, thus comparing our results with other studies become difficult. Even that, we are confident that our work provides key insight and information about the environmental significance of nitrogen and phosphorus regeneration as mediated by marine copepods.

One of the difficulties in our experimental studies is simulating the in situ conditions in a more realistic way. Research on zooplankton, especially on copepods, commonly use experimental densities much higher than those encountered in the field. In our case, since the nutrient concentrations in the study area were quite low, particularly for ammonium, it was necessary to concentrate the number of copepods in incubations. We have now deeper discussed this issue. Although we are aware that our results could be overestimated, our

conclusions are also supported by a set of previous works in the field (Perez-Aragon et al., 2011; Valdes et al., 2017; Valdes et al., 2017).

2) SPECIFIC COMMENTS

2a) Introduction:

p.2 lines 20, 25, 26. Copepods may excrete much more than 53% of their body nitrogen in the form of ammonia and this percentage is highly variable (as said in line 25). There are many papers on this subject, including review papers.

We agree. We have now provided a better assessment of the subject and improve citations.

p.3 lines 16-17: How is the presence of an oligotrophic gradient supported? Nutrient values from the only three stations are insufficient. Nutrient and/or Chla data from more stations would be helpful.

This paragraph was rephrased. An explanation about the gradient selection and its support was given in detail in the Methods section. Briefly, the three LD stations were chosen along a regional gradient in oligotrophy and they were selected using satellite imagery, altimetry and Lagragian diagnostic (Moutin et al., 2017). A complementary approach included the abundance of selected diazotrophs *nifH* gene copies (on board; Stenegren et al., 2017).

In the revised text, we removed the oligotrophic gradient from the description of the study area and now focus only in the general study area: WTSP. Details on how the oligotrophic gradient was determined was added to the Methods section.

2b) Methodology:

General remark for the experimental set up: an additional control with copepods only would have being very helpful.

p.3 line 22: please define "long duration"

We have now defined long duration stations in Material and Methods section.

p.3 line 22: please add coordinates of the stations

Coordinates were added. Additionally, we have now added a map of the transect during the cruise with the stations.

p.3 lines 24-25: please add maximum and minimum values of chlorophyll-a

Details about the chlorophyll values and about sampling strategy during OUTPACE cruise are available in Moutin et al., 2017. We have now provided the reference for this in Methods section as well asthe chlorophyll-a values in the DCMin Table 1, which is the depth in which we took the seawater for our experiments.

p.4 lines 2-4: please specify if tow was vertical (or oblique), tow speed, net diameter

We have now provided these sampling details in "Mesozooplankton sampling" section.

p.4 line 13: the field composition of copepods and other zooplankton should be also shown in Table S1

More details about the compositions of copepods and other zooplankton in the field were provided in Carlotti et al., (this issue), andthe abundance of copepods in the study area was provided in the second paragraph of Discussion section.

p.4 lines 23-24: please specify % of mortality

We have now provided the percentage of mortality during the incubations in the "Preparation of the microcosm and experimental setup" section.

p.8 lines 21: check that Redfield ratio of organic nutrients is < 16:1

Results were checked and modified accordingly.

p.9 lines 7-8: unclear meaning

These lines were clarified.

2c) Results:

General remark. Do not repeat values that can be found in tables or figures unless necessary.

Table 1: use either 2 or 3 decimals depending on the precision of the method for each parameter

Done.

Table 1: add Temperature, Chla, DON, DOP, values at DCM

The information was added. DON-DOP and Chla were extracted from Moutin et al., (this issue) and from Dupoy et al., (this issue), respectively.

p.7 line 25: "first treatment", replace by "treatment with copepods"

Done.

p.8 line 4: "significant difference in time,". Please add results of statistical test

We added the information to the text.

2d) Discussion:

General remark. Do not repeat detailed description of results, but only briefly giving outcome in connection with related literature.

p.10 line 15 and line 17: Table 1 shows that MA stations (LD A & LD B) are characterized by higher inorganic nitrogen but not higher inorganic phosphorus concentration than SG (LD C).

The information was checked and corrected accordingly.

p.10 line 25: "influence on biogeochemical variability": not clear.

We have now clarified this section in the text.

p.10 lines 31-32: Since there was no significant difference between treatments in LD A for bacterioplankton abundance (p.8 lines 26-27), a conclusion on an effect of zoo- plankton cannot be made.

This issue was discussedin the discussion section.

p.11 lines 7 to 10. LD B is not in the South Pacific Gyre (p.1 line 24). Check and eventually move this part to discussion on LD C which is in the South Pacific Gyre.

We have now restructured this section to make it pertinent to the station sampled.

p.13 lines 33-34: Too general conclusion on the role of zooplankton metabolism, not supported by manuscript's observations.

We have now rewritten our conclusion, which focuses more on our observations.

2e) Bibliography:

References to articles in preparation should be avoided

We modified the text accordingly.

3) TECHNICAL CORRECTIONS

p.2 line 2, p.8 line 18, p.10 line 12, p.10 line 15, p.12 line 32, p.13 lines 8-9: English grammar mistakes

These sections have been revised and modified accordingly.

p.2 line 10: change order of references

We have now modified the order of the references.

p.3 line 3: ". . .in a phosphorus limited. . ."

Corrected.

p.3 line 5: ". . .Gasol, 2007"

Corrected.

p.3 line 8: ". . .zooplankton-produced DOM by bacterial communities. . .": unclear expression

We have now clarified this expression.

p.4 line 14: "acclimation" instead of "acclimatization"

Corrected.

p.4 line 21: "such that 6 bottles were added in that case": not clear

We have now clarified this sentence.

p.5 line 23: correct "u sing"

Corrected.

Please correct typing mistakes and journal names abbreviations in bibliography

Done.

RC2

The authors present a study that simultaneously examines the role of zooplankton and bacteria on nitrogen and phosphorus cycling in low-nutrient systems. Few studies have tried to assess these factors simultaneously to link the two processes. Although the authors provide a lot of interesting results, I am not convinced that their data back up the larger assertions of the role of copepod excretory products play in spurring growth of microbes. Also, I am not convinced that in situ conditions were accurately enough recreated to directly apply these results to the in situ setting.

More specifically:
1) It is challenging to know how much N and P is being utilized by the bacteria (and in what forms) when you don't have baseline data for N and P excretion rates for copepods in the absence of microbes. Without this information, you are unable to say the actual impact either party (copepods or microbes) is having on buildup or drawdown of different N or P compounds.

2) In many cases, there were not significant or clear differences between the controls and copepod treatments. For instance, the bacterial community structure for LD A and LD B stations grouped more by time point than by treatment. That implies an effect of the incubation itself, and not due to extra nutrients brought on zooplankton excretion. Also, in 2 out of the 3 experiments, it appears that bacterial abundance was higher in the controls at the end of the experiment. If they were using the nutrients to grow, we would expect to see significantly higher abundances in the copepod bottles.

3) Copepod densities in the bottles were much higher than in situ conditions. While high densities of zooplankton are definitely necessary for detectable changes in excretory products, this also meant that the change in N and P facilitated by zooplankton is likely to be much larger within this experiment than what bacteria experience in the field.

4) When looking at the dendrogram results in Figure 8, it appears that the DNA samples taken from the in situ environment show a very different community structure when compared to the communities that were present in the incubation bottles at time T0. If the community structure in the bottles at the beginning of the experiment isn't representativeof the local community, the changes observed might not reflect what would occur in situ.

5) I am concerned that there are significant differences between copepod and control treatments at T0. The methods say that the organisms were placed in new incubation water at the beginning of the excretion step. I assume initial samples were taken immediately.While the authors attempt to explain the differences seen in DON (p. 11, line 26), there is no information addressing the differences in bacterial community structure at T0 for several of the experiments.

Additionally, the entire document needs to be reviewed for use of proper English grammar. Most of the issues are minor, but do make the paper a bit harder to read and understand. I will point out several below in my comments, but there are certainly more within the document.

Attending to the general comments:
   (1) It is true that the "microbe free" effect of copepod excretion on free-living microorganism's metabolism is difficult to address and is certain not the scope of our studies. We consider the copepod as a hole considering also its own microbiome and potential microbial epibionts. Have a "microbe free" experiment is also not realistic itself if we try to determine nutrient fluxes and potential role of copepod. Our approach is based on a series of previous studies about the excretion of copepods and its impact on microbial communities (Perez-Aragón et al., 2011; Valdés et al., 2017a; Valdés et al., 2017b).We have a robust data base regarding the type of compounds excreted by copepods in central-southern Chile. In addition, the excretion of copepods has been studied for several decades and there is an important amount of works and reviews about this topic (Bidigare, 1983; Ikeda et al., 2001; Steinberg and Saba 2008; Hernandez-León et al., 2008; Smith and Whitledge 1977; Peréz-Aragón et al., 2011). Several of these studies have demonstrated that the main compounds excreted by copepods are DON, ammonium. and DOP, although the latter has been much less studied than nitrogen compounds.

   (2) We agree with the reviewer that incubation time was an important factor and that the enrichment associated with copepods could be evidenced mainly at the beginning of the incubation, since a later bacterial response could be associated with microbial metabolisms. However, our results indicate that presence of copepods and its metabolisms end products enhance the activity of different microbial communities in very short time response (<2h). We identify examples of active

bacteria in our incubations such as LDA, known to avidly react to the different compounds potentially provided by copepods. References associated, for example, to Alteromonadales were included in the discussion of the manuscript. Moreover, the reviewer can check for example in Figure 10 that different active microbial communities were observed in all the stations between the treatment and controls. Therefore, the effect of the copepod metabolism can´t be discarded either. The fact that this "activity enhancement" did not result in significant increase in abundance also indicates that copepods interaction with bacterioplankton could be more complex than expected potentially generating shifts among microbial groups, since some bacteria presented a reduction in the copepod treatments.

(3) We agree in this point, and also this issue was taken in consideration in the Discussion section.

(4) Figure 8 shows the RNA samples not DNA, and thus the active bacterioplankton community. The most abundant communities present in situ and during the experiments are the same, the changes respond to its contribution. Here we study the activity of the microbial communities and as explained in the point (2) has been reported that some opportunistic microorganism as Gammaproteobacteria increase their activity in a very short time in response to enrichment experiments.DNA in situ was used to identify groups present in the study area which could however not usually be the most active ones. But T0 in the graphs were obtain from the activity of bacterioplankton using RNA so is not unexpected to have a different composition particularly in the case of cyanobacteria or photoautotrophs. Moreover, its common that the incubation experiments the microbial communities differs (in diversity and composition) from the field, since the incubation s may favor microorganism rare in nature but featuring opportunistic that allow for a more rapid adaptation to changes in environmental conditions and hence, outcompete other that are originally more abundant. This explanation and references were included in the Discussion section.

(5) The initial samples were taken once all the bottles were incubated both with or without copepods. The time delay between the first bottle put in incubation and the beginning (time T0) of the experiments was around 20 minutes. This time window can explain the differences found in DON concentrations at the initial sampling points and also the variations in contribution of the most abundant groups. This information has been now added in Methods section.

Specific Comments:

Introduction:
*P. 2, line 7-8: Citation needed for sentence about thermal stratification and global warming decreasing the nutrient supply
We have now provided the proper reference.

*P. 2, lines 19-23. There has been a lot of work on zooplankton excretion and the various products. A more comprehensive summary should be provided about N cycling of body nitrogen and support for phytoplankton needs in oceanic environments.

We have now added a more comprehensive summary, with new references.

*P. 3, line 1: "enhance" should be "enhances."
Corrected.

Methods:
*P. 3. Second sentence of section 2.1: Should read "The transect began west of New Caledonia and ended near Tahiti."
Corrected.

*P. 3, section 2.1: You should include latitude/longitudes for your LD stations. Also, you should include a map of the transect and your specific sites used for the LD experiments.
Latitude and longitudes were now included in the first section of methods along with a map with the transect during the cruise and stationssampled.

*P. 3, line 26: it says that you used a CTD rosette to collect water. Further information on hydrographic parameters should be included. Especially on chlorophyll concentrations within the DCM.
Temperature and Chl-a values have been included in the Table 1 on the new version of the MS.

*P. 4, line 7: was the incubation seawater also collected from the DCM? I assume so, but that is not clearly stated.
Yes, the incubation seawater was obtained from the DCM. We have now clarified this in section 2.3 of the new version of the MS.

*P. 4, line 8: isn't the acclimatizing step number 1?
Yes, acclimation step is number 1. We have now corrected.

*P. 4, line 8-9: If you were using the filtered seawater, it should read "seawater (22.5 L) was immediately filtered through a 0.7 um filter…."
Corrected.

*P. 4, line 10: 25C doesn't really seem cold. Perhaps "temperature controlled room" is a better term than "cold room?"
Thanks for the observation, we have now modified to "temperature controlled room".

*P. 4, line 18: by controlled temperature, is that also in the 25 degrees C room? You should specifically say the temperature.
We have now provided temperature values throughout the MS.

*P. 4, line 21: what 6 bottles were added? Were they all controls? You need to explain this furtherâ˘Aˇ Tit is confusing as currently written.
We have now clarified this in "preparation of the microcosm and experimental setup" section.

*P. 4, line 24: should be "to ensure" instead of "of ensuring"

Corrected.

*P. 4, line 26: suggest specifying "During the excretion phase…"
We agree, thanks for the suggestion.

*P. 5, line 12: I am not familiar with the wet oxidation method, so I looked up the paper you cited. It looks that paper assesses PON and POP. What modifications need to be made to look at dissolved materials?
The wet oxidation method is used for both particulate and dissolve compounds, the differences is that the oxidation procedures uses seawater samples previously filtered through pre-combusted GF/F filter in Teflon flask for dissolved face and for the particulate sample, the filter is used in the Teflon flask for the oxidation procedures. Details are provided in "dissolved inorganic and organic measurements" section.

*Sections 2.5-2.6: My expertise is in zooplankton physiology, so I cannot assess the suitability of the genetics and bacterial methods.

*P. 5, line 22: "u sing" should be "using"
Corrected.

Results:
*Generally speaking, the authors did a good job displaying a very complicated data set. However, there are some places where the clarity of the results presentation can be improved with some minor modifications. I provide some of those suggestions below for Table S2 and Figures 2-5 and 8.

*P. 7, line 21: If these are ANOVA results, you should say so.
Done for the entire Results section.

*P. 7, line 24: "notorious" is not a correct term to use. "Notable" instead?
We agree, we have now modified.

*P. 7, line 30: I don't understand why there is a difference at time T0. Is there a gap between when you started the experiment and took your initial sample? If that is the case, you need to specifically mention that in the methods.
Yes, there is a gap between the moment we started the experiments and the moment we took the initial sample. The initial sample (T0) was taken after all the bottles were conditioned (when all the bottles had the copepods and/or the seawater). The gap of the time was around of 15-20 minutes. We have now added this information in Methods section.

*P.7, line 30-31: Your final sentence appears to contradict itself. If there were nonsignificantdifferences, wouldn't the p-value need to be >0.05?
We have now corrected. Thanks for the observation.

*P. 8, line 1-2: Consider re-wording for clarity. Specifically, all of the statistical test results should be together within parentheses.

We have now clarified this sentence and all the statistical tests were in parentheses.

*P. 8, line 6: missing a "(" before the "LD B: : :"
Corrected.

*P. 8, line 7: "trough" should be "through"
Corrected.

*P. 8, line 8: "estimated" does not make sense here. Perhaps "significant differences were only observed between: : :"
Thanks for the observation, now it was modified accordingly.

*P 8, second paragraph:
(1) You should be very specific here about which N:P ratio you are discussing at any given time. It is confusing at the beginning when you define it only using nitrate/nitrite and phosphate, but then discuss inorganic and organic ratios.
We modified the nomenclature to DIN:DIP and DON:DOP for inorganic and organic ratios, respectively.

(2) Would you expect DON and DOP to be comparable to Redfield? Urea (one of the more common forms of DON), actually has 2 N per molecule, so if it is molar ratios that you are using, then it is not a 1:1 comparison that could be made between DON and DOP. Are there other papers that have using the organic N:P ratio? If so, it would be really helpful to bring those in to support your use and interpretation of this ratio.
We did not measured urea, so it is not the molar ratio that we are using. DON: DOP ratio is much less commonly measured than the inorganic ratio and only we have a couple of references with measure of this ratio to compare our interpretation.

(3) Line 18: you say that the DON:DOP is close to Redfield. But, the numbers that you mention in the prior sentences (and those in figure 4) are much higher than Redfield.
We agree and now it was modified.

(4) Line 20-21: The deviation from Redfield appears to be that LD C is much higher, not lower.
We agree and now it was modified.

(5) Line 21: should be "ratio" not "ration"
Corrected.

*P. 8, line 23: should be "shown" not "showed"
Corrected.

*P. 8, line 24: "plankton" is a plural term, so it should be "bacterioplankton were more abundant…"
Done.

*P. 8, line 30: is "and between time" referring to a different time than the sampling times

you mention earlier in the sentence?
Yes, we have now deleted the sentence.

*P. 9 , line 14: On the other hand makes it sound like this is contradictory result. The two genetic markers seem to be showing similar trends. "Additionally" might be a better way to begin this thought.
Agree. Now it was modified.

*P. 9, line 27: if LD A and LD B show variability associated with time points, can you draw conclusions then based on presence of copepods? Or is this an artifact of the culturing conditions?
This issue was discussed in the general comment (2).

*P. 10, line 2: do you mean "larger" instead of "longer?"
Yes. We have now modified.

*P. 10, line 12: "show" instead of "showed"
Corrected.

Discussion:
*Section 4.0: Can you say the MA region had higher nutrient concentrations when LD B had the lowest of all of your sites?
The section 4.0 have been now modified, avoiding a general analysis of both regions.

*P. 10, line 22: "region" should be "regions"
Corrected.

*P. 10, line 27: You said there was significant increase in DON. But this was only during one time point. The other one is during timepoint T0, which seem suspicious.
The increase in DON at different time or in only one-time point during the experiments has been reported by other authors (Gardner and Paffenhöfer, 1982;Steinberg et al., 2002). DON excretion is release in a non-continuous way in highly concentrated spurts, while ammonium is release continuously, which is in agreement with our results.

*P. 10, line 31: do you mean "increase" instead of "increment?"
Yes. Thanks for the observation.

*P. 10, line 31: is 22,8x10^3 correct? Should it be 228 or 22.8? The formatting is a bit odd and makes me think it is a typo.
The formatting is correct. $22.8 \times 10^3 = 22,800$.

*P. 10, line 32: You suggest DON assimilated by growing bacteria. But it is mostly in the control of LD A that the bacterial abundances increase between 1-2 hours.
In average, bacteria increase their abundance but between 1 and at the end of incubation. We have now modified this sentence.

*P. 11, line 10: But in LD B, you said no significant differences were observed over time

(although your reported stats results don't match that - p 7, line 30)
We agree. We have now deleted this sentence.

*P. 11, line 26: How long after adding the copepods did you take your initial measurements?Could it really happen that fast? You need some more specific information toback up this as a reasonable assertion.
The initial sample was taken after all the bottles set for incubation (T0), that is when all the bottles had the copepods and/or the seawater. The gap of the time was around of 15-20 minutes. We have now added this information in methods section.

*P. 11, line 27: "crustaceans" instead of "crustacean"
Corrected.

*P. 12, line 10: Is this supposed to be phosphate or DOP? All the rest of the P references in this paragraph are to DOP, so I wanted to double-check.
Thanks for the observation, now it has been corrected

*Section 4.2, first paragraph:
(1) But it didn't seem like bacterial abundance was increased within the copepod bottles for most of the timepoints.
(2) You said in your bacterial community results that LD B experiments grouped by time primarily, not by treatment.
(1)(2)We agree and we have now rewritten this section.

*P. 13, line 18: "diminished" instead of "diminish"
Ok, corrected.

*P. 13, line 28: Define "substantial amounts." You need to be able to better quantify this. I don't really think this statement is firmly backed up by the data presented
We have now delete "substantial amounts" to avoid misunderstanding. However, our results showed an increase in DOP concentrations at different sampling points during the experiments.

*P. 13, lines 32-34: It didn't really seem like you saw bacterial growth overall. This statement is not directly supported by your results.
We agree, we have now modified this sentence.

Tables and Figures:
*Table 1: Redefine what this N:P ratio includes in the table caption.
We have now clarified in the Table 1, Figure 4 and through the whole manuscript.

*Figure 1:
(1) You mention "in situ temperature" a few times. What is that temperature?
We have now provided temperature values at DCM in Table 1.

(2) What does "*" mean after the 0.25 sampling point?

"*" correspond to the additional sampling time that we took at LD B and LD C stations. We have now clarified in the figure caption.

(3) In the caption, it says the feeding phase lasts for 3 hours. But other places in the document it states 4 hours.
Thanks for the observation. Now it was clarified.

(4) Specify in the caption that each of the 15 bottles contains 10 fed copepods.
Done.

*Table S2: You should include the overall ANOVA results as part of Table S2. It would be much easier for the reader if we could find all of those statistical results in one place. Perhaps on the gray lines?
We try to add all the information in one table, but is difficult to follow the results and is necessary extend the table losing clarity. Thus, we decided not added the information in the Table S2.

*Figures 2-5: The data are really complicated and results are hard to grasp between the text (where ANOVA are reported), table S2 (where post-hoc data are reported), and graphs (where visual trends are reported). Can you find a way to mark statistically significant differences on these graphs? Then the reader can go to these for a summary.
We tried but it's hard to put all these information in the graphs without lose clarity in the figure, since we should add this information between sampling time and treatments. Sorry for this thusdo not agreed with adding the statistical information in the figures.

*Figure 8: Can you use the prefixes in the descriptions of each line? "T1 copepod" or "T2 Control." It makes it very difficult to see the trends you mentioned when you have to look back at the legend each time you read a line to figure out the color-coded timepoint.
We agree. Now it was modified in the figure.

[revised manuscript text omitted]

| Página 8: [1] Eliminado | Valentina Valdes Castro | 10/4/18 12:30:00 |
|---|---|---|

notorious

| Página 8: [1] Eliminado | Valentina Valdes Castro | 10/4/18 12:30:00 |
|---|---|---|

notorious

| Página 8: [1] Eliminado | Valentina Valdes Castro | 10/4/18 12:30:00 |
|---|---|---|

notorious

| Página 8: [1] Eliminado | Valentina Valdes Castro | 10/4/18 12:30:00 |
|---|---|---|

notorious

| Página 8: [1] Eliminado | Valentina Valdes Castro | 10/4/18 12:30:00 |
|---|---|---|

notorious

| Página 8: [1] Eliminado | Valentina Valdes Castro | 10/4/18 12:30:00 |
|---|---|---|

notorious

| Página 8: [1] Eliminado | Valentina Valdes Castro | 10/4/18 12:30:00 |
|---|---|---|

notorious

| Página 8: [1] Eliminado | Valentina Valdes Castro | 10/4/18 12:30:00 |
|---|---|---|

notorious

| Página 8: [1] Eliminado | Valentina Valdes Castro | 10/4/18 12:30:00 |
|---|---|---|

notorious

| Página 8: [1] Eliminado | Valentina Valdes Castro | 10/4/18 12:30:00 |
|---|---|---|

notorious

| Página 8: [2] Eliminado | Valentina Valdes Castro | 13/4/18 13:35:00 |
|---|---|---|

| Página 8: [2] Eliminado | Valentina Valdes Castro | 13/4/18 13:35:00 |
|---|---|---|

| Página 8: [2] Eliminado | Valentina Valdes Castro | 13/4/18 13:35:00 |
|---|---|---|

| Página 8: [2] Eliminado | Valentina Valdes Castro | 13/4/18 13:35:00 |
|---|---|---|

| Página 8: [2] Eliminado | Valentina Valdes Castro | 13/4/18 13:35:00 |
|---|---|---|

| Página 8: [2] Eliminado | Valentina Valdes Castro | 13/4/18 13:35:00 |
|---|---|---|

| Página 8: [2] Eliminado | Valentina Valdes Castro | 13/4/18 13:35:00 |
|---|---|---|

| Página 8: [3] Eliminado | Valentina Valdes Castro | 16/4/18 18:10:00 |
|---|---|---|

N:

| Página 8: [3] Eliminado | Valentina Valdes Castro | 16/4/18 18:10:00 |
|---|---|---|

N:

| Página 8: [3] Eliminado | Valentina Valdes Castro | 16/4/18 18:10:00 |
|---|---|---|

N:

| Página 8: [3] Eliminado | Valentina Valdes Castro | 16/4/18 18:10:00 |
|---|---|---|

N:

| Página 8: [3] Eliminado | Valentina Valdes Castro | 16/4/18 18:10:00 |
|---|---|---|

N:

| Página 8: [3] Eliminado | Valentina Valdes Castro | 16/4/18 18:10:00 |
|---|---|---|

N:

| Página 8: [3] Eliminado | Valentina Valdes Castro | 16/4/18 18:10:00 |
|---|---|---|

N:

| Página 8: [3] Eliminado | Valentina Valdes Castro | 16/4/18 18:10:00 |
|---|---|---|

N:

| Página 8: [3] Eliminado | Valentina Valdes Castro | 16/4/18 18:10:00 |
|---|---|---|

N:

| Página 8: [3] Eliminado | Valentina Valdes Castro | 16/4/18 18:10:00 |
|---|---|---|

N:

| Página 8: [3] Eliminado | Valentina Valdes Castro | 16/4/18 18:10:00 |
|---|---|---|

N:

| Página 8: [3] Eliminado | Valentina Valdes Castro | 16/4/18 18:10:00 |
|---|---|---|

N:

| Página 8: [3] Eliminado | Valentina Valdes Castro | 16/4/18 18:10:00 |
|---|---|---|

N:

| Página 8: [3] Eliminado | Valentina Valdes Castro | 16/4/18 18:10:00 |
|---|---|---|

N:

| Página 8: [3] Eliminado | Valentina Valdes Castro | 16/4/18 18:10:00 |
|---|---|---|

N:

| Página 8: [3] Eliminado | Valentina Valdes Castro | 16/4/18 18:10:00 |
|---|---|---|

N:

| **Página 8: [3] Eliminado** | **Valentina Valdes Castro** | **16/4/18 18:10:00** |

N:

| **Página 8: [3] Eliminado** | **Valentina Valdes Castro** | **16/4/18 18:10:00** |

N:

| **Página 8: [3] Eliminado** | **Valentina Valdes Castro** | **16/4/18 18:10:00** |

N:

| **Página 8: [3] Eliminado** | **Valentina Valdes Castro** | **16/4/18 18:10:00** |

N:

| **Página 8: [3] Eliminado** | **Valentina Valdes Castro** | **16/4/18 18:10:00** |

N:

| **Página 8: [3] Eliminado** | **Valentina Valdes Castro** | **16/4/18 18:10:00** |

N:

| **Página 8: [4] Eliminado** | **Valentina Valdes Castro** | **24/4/18 11:00:00** |

ed

| **Página 8: [4] Eliminado** | **Valentina Valdes Castro** | **24/4/18 11:00:00** |

ed

| **Página 8: [4] Eliminado** | **Valentina Valdes Castro** | **24/4/18 11:00:00** |

ed

| **Página 8: [4] Eliminado** | **Valentina Valdes Castro** | **24/4/18 11:00:00** |

ed

| **Página 8: [4] Eliminado** | **Valentina Valdes Castro** | **24/4/18 11:00:00** |

ed

| **Página 11: [5] Eliminado** | **Valentina Valdes Castro** | **17/4/18 11:07:00** |

| **Página 11: [6] Eliminado** | **Valentina Valdes Castro** | **16/3/18 12:28:00** |

| **Página 11: [7] Eliminado** | **Valentina Valdes Castro** | **16/3/18 16:50:00** |

In addition, the highest differences in DON concentration between the treatment with copepods and the control (without copepods) were found at initial time points, mainly in LD A and LD C experiments. Zooplankton can excrete DON in highly concentrated spurts lasting up to an hour, unlike ammonium which is excreted continuously by crustacean (Gardner and Paffenhöfer, 1982; Steinberg et al., 2002). In this study, we cannot attribute the increase in the concentration of the different compounds only to the excretion process since we did

not monitor possible leaching from fecal pellets produced during the experiments. The leaching is also one of the possible pathways through which zooplankton can release dissolved compounds, along with the sloppy feeding and excretion. However, it has been demonstrated that the contribution provided by leaching is insignificant compared to excretion, less than 6% in the case of the release of DON, whereas ammonium was undetected (Saba et al., 2011; Steinberg et al., 2000). Thus, if we consider leaching from fecal pellets as negligible, then DON accumulation could account for up to 1.91 µmol $L^{-1}$ $h^{-1}$ excretion in the LD A experiment. However, the experimental copepod densities were several orders of magnitude higher than those expected in situ (10 copepods for 0.5 L volume, i.e. 20,000 ind. $m^{-3}$), so that our results might be overestimated because copepods concentration in our bottles did not reflect the in situ abundance (MA: 1,500 ind. $m^{-3}$ and SG: 500 ind. $m^{-3}$; Carlotti et al., this issue). However, such concentration was strictly necessary through the experiments to obtain a signal in ammonium concentrations, which were very low in the study area (Table 1). In addition, we cannot estimate excretion rates for all compounds since they appear later in the incubation probably modulated by microbial metabolism.

| Página 11: [8] Eliminado | Valentina Valdes Castro | 16/3/18 16:58:00 |
|---|---|---|

In LD A experiments, phosphate was found in higher concentrations in the treatment with copepods compared to the control between 0 and 1 h of incubation. In LD B and LD C experiments, concentrations in the treatment with copepods versus control were significantly higher at the beginning and at the end of the incubation for LD B, also at the end of incubation for LD C.

| Página 12: [9] Eliminado | Valentina Valdes Castro | 16/4/18 12:51:00 |
|---|---|---|

, even if the abundance was slightly increased in the experiments with copepods. In our LD B and LD C experiments, this impact on bacterial community structure was significantly different with additions of copepods compared to the control.

Through the incubations, the bacterioplankton community was characterized by changes

| Página 13: [10] Eliminado | Valentina Valdes Castro | 16/3/18 17:33:00 |
|---|---|---|

| Página 13: [11] Eliminado | Valentina Valdes Castro | 17/4/18 12:05:00 |
|---|---|---|

| Página 13: [12] Eliminado | Valentina Valdes Castro | 14/3/18 17:30:00 |
|---|---|---|

seems to play a critical role in nutrient recycling by fuelling microbial and phytoplankton growth in the vast oceanic regions of the world ocean.

| Página 18: [13] Eliminado | Valentina Valdes Castro | 17/4/18 12:17:00 |
|---|---|---|

1. Copepod acclimation

Incubated 10 copepods
at in situ temperature

x15

After 4-6 h

Transferred
copepods

2. Copepod feeding

x15

After 4 h

Transferred
copepods

Natural
bacterioplankton
assemblages
from DCM
FSW 0.7 μm

3. Copepod+microbial recycling

✓ Sampling points: 0, 0.25*, 0.5, 1, 2 and 4 h
✓ Samples for: Nutrients ($NO_3^-$, $NO_2^-$ and $PO_4^{3-}$)
    DON-DOP
    $NH_4^+$
    Bacterioplankton abundance
    RNA sample

Control

X 3
Incubated at *in situ* temperature

[Figure]

**(a)**

**(b)**

**(c)**

*Time*
- T0 (green triangle)
- T1 (blue inverted triangle)
- T2 (cyan square)
- T3 (red diamond)
- T4 (magenta circle)
- Initial (grey plus)

Legend:
- Acidobacteria
- Actinobacteria
- Bacteroidetes
- Chloroflexi
- Cyanobacteria
- Firmicutes
- Marinimicrobia (SAR406 clade)
- Nitrospinae
- PAUC34f
- Planctomycetes
- Poribacteria
- Alphaproteobacteria
- Betaproteobacteria
- Deltaproteobacteria
- Gammaproteobacteria
- Verrucomicrobia
- Other (<0.5 %)

Relative abundace (%)